# SPEED-Bench: A Unified and Diverse Benchmark for Speculative Decoding

Talor Abramovich [* 1]   Maor Ashkenazi [* 1]   Izzy Putterman [1]   Benjamin Chislett [1]   Tiyasa Mitra [1]
Bita Darvish Rouhani [1]   Ran Zilberstein [1]   Yonatan Geifman [1]

## Abstract

Speculative Decoding (SD) has emerged as a critical technique for accelerating Large Language Model (LLM) inference. Unlike deterministic system optimizations, SD performance is inherently data-dependent, meaning that diverse and representative workloads are essential for accurately measuring its effectiveness. Existing benchmarks suffer from limited task diversity, inadequate support for throughput-oriented evaluation, and a reliance on high-level implementations that fail to reflect production environments. To address this, we introduce **SPEED-Bench**, a comprehensive suite designed to standardize SD evaluation across diverse semantic domains and realistic serving regimes. SPEED-Bench offers a carefully curated *Qualitative* data split, selected by prioritizing semantic diversity across the data samples. Additionally, it includes a *Throughput* data split, allowing speedup evaluation across a range of concurrencies, from latency-sensitive low-batch settings to throughput-oriented high-load scenarios. By integrating with production engines like vLLM and TensorRT-LLM, SPEED-Bench allows practitioners to analyze system behaviors often masked by other benchmarks. We highlight this by quantifying how synthetic inputs overestimate real-world throughput, identifying batch-size dependent optimal draft lengths and biases in low-diversity data, and analyzing the caveats of vocabulary pruning in state-of-the-art drafters. We release SPEED-Bench to establish a unified evaluation standard for practical comparisons of SD algorithms.[1]

## 1. Introduction

Large Language Model (LLM) inference has become a cornerstone of modern AI applications, yet it remains fundamentally bottlenecked by the autoregressive decoding process. On modern hardware, this decoding phase is predominantly memory-bound in low-concurrency settings: the time required to move model parameters from High-Bandwidth Memory (HBM) to the GPU's on-chip caches far exceeds the time spent on actual computation. While techniques like KV-Caching mitigate recomputation costs, they leave the GPU's compute units significantly underutilized during the sequential generation of tokens. This inefficiency provides the primary motivation for Speculative Decoding (SD) (Leviathan et al., 2023; Chen et al., 2023). By using a relatively small "draft model" to speculate a sequence of future tokens, the system can perform a single "verification" forward pass using the larger target model, effectively generating multiple tokens. Because memory reads dominate latency in standard decoding, processing multiple tokens simultaneously incurs only a marginal overhead compared to a single token and significantly improves throughput, provided accurate speculations. Notably, frontier models such as DeepSeek-R1 (Guo et al., 2025), Qwen3-Next (Yang et al., 2025a), Nemotron-3 (Blakeman et al., 2025), and MiMo-V2-Flash (Xiaomi, 2026) have natively integrated Multi-Token Prediction (MTP) heads into their architecture to support efficient drafting. When combined with rejection sampling, SD remains a lossless acceleration method, exactly matching the output distribution of the target model.

Despite its rapid adoption, the evaluation of SD algorithms remains fragmented and often unrepresentative of real-world environments. We identify several gaps in current literature. Firstly, draft acceptance rates are highly sensitive to data domain and entropy, yet new methods are frequently validated on inconsistent datasets, making cross-method comparison difficult. Furthermore, popular datasets such as MT-Bench (Zheng et al., 2023) suffer from limited prompt volume and a lack of intra-category diversity, failing to capture the complexity of real-world data distribution. Secondly, papers often evaluate methods using high-level libraries, such as HuggingFace (Wolf et al., 2020), that do not reflect the additional optimizations found in production engines like vLLM (Kwon et al., 2023), TensorRT-LLM (NVIDIA,

---

[*]Equal contribution   [1]NVIDIA. Correspondence to: Talor Abramovich <talora@nvidia.com>, Maor Ashkenazi <mashkenazi@nvidia.com>.

*Proceedings of the $43^{rd}$ International Conference on Machine Learning*, Seoul, South Korea. PMLR 306, 2026. Copyright 2026 by the author(s).

[1]Our data is on 🤗 HuggingFace.

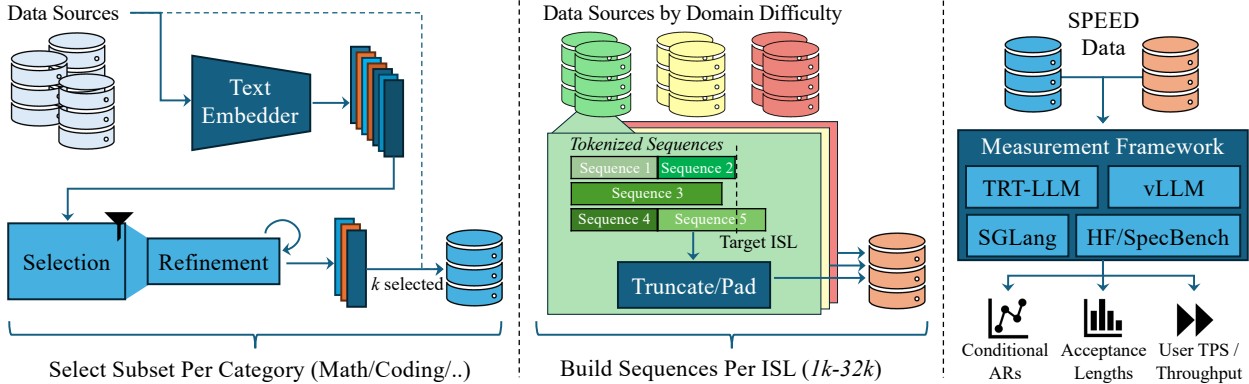

*Figure 1.* **Overview of the SPEED-Bench ecosystem. (Left)** Curation of the *Qualitative split*, utilizing a custom selection algorithm on prompt embeddings to maximize semantic diversity across categories. **(Middle)** Construction of the *Throughput Split*, where data is aggregated and processed into fixed Input Sequence Length (ISL) buckets (1k-32k) across three domain difficulties, supporting large batch sizes (up to 512 per ISL and difficulty). **(Right)** The unified measurement framework used to report standard SD metrics and speedups.

2023), or SGLang (Zheng et al., 2024). Thirdly, research often focuses on $BatchSize(BS) = 1$ to report speedups. However, real-world multi-user model serving prioritizes throughput, which is usually optimized using larger batches; this shifts the inference process toward a compute-bound regime, often diminishing the speedups observed with SD. Finally, existing benchmarks predominantly feature short Input Sequence Lengths (ISL). As the industry shifts towards long-context applications like coding assistants, this long ISL regime remains largely unstudied.

To bridge these gaps, we introduce SPEED-Bench (**Spe**culative **E**valuation **D**ataset). Figure 1 illustrates the three components of SPEED-Bench: 1) a *Qualitative* data split optimized for semantic coverage across categories, 2) a *Throughput* data split tailored for measuring speedups under realistic model serving scenarios, and 3) a unified measurement framework. This design allows evaluating SD across varying text domains, ISLs, and concurrencies using production-grade inference engines. We further provide integration with SpecBench (Xia et al., 2024) models to support easier evaluation for the research community. We summarize our main contributions are as follows:

**1) The SPEED-Bench Dataset:** A multipurpose dataset designed for both qualitative analysis of draft accuracy across diverse categories, and throughput measurements of realistic workloads across varying ISLs and concurrencies, all without the overhead of massive-scale testing.

**2) Measurement Framework:** A unified evaluation pipeline integrated with production-grade engines, supporting measurements of real-world speedups.

**3) Empirical Analysis:** We demonstrate the utility of

SPEED-Bench by analyzing properties and side effects often masked by traditional benchmarking methodologies.

**Conflict of Interest Disclosure** The authors are employed by NVIDIA, which leads the development of TensorRT-LLM and two of the draft models evaluated in this paper.

## 2. Related Work

**Speculative Decoding** The efficiency of SD relies on balancing drafter accuracy and latency. Early *Vanilla SD* approaches used smaller, standalone models for drafting (Leviathan et al., 2023; Chen et al., 2023). To reduce the memory and compute overhead of running a separate model, recent methods integrate drafting directly into the target model: *Medusa* (Cai et al., 2024) and *EAGLE* (Li et al., 2024a;b; 2025b) attach lightweight drafting heads, while frontier models like Qwen3-Next (Yang et al., 2025a) adopt *Native Multi-Token Prediction (MTP)* to eliminate post-trained modules entirely. Other work focused on optimized speculation strategies. Tree-based methods verify multiple branches simultaneously to maximize acceptance (Miao et al., 2024; Chen et al., 2024), and alternative approaches generate tokens in parallel rather than autoregressively (An et al., 2025; Xiao et al., 2024). *MagicDec* (Sadhukhan et al.) and *LongSpec* (Yang et al., 2025b) focus on long context scenarios, tackling accuracy drop over extended sequences. Finally, N-grams[2] and lookahead strategies (Fu et al.; He et al., 2024) utilize training-free drafting heuristics.

**Benchmarking Methodology** Evaluation of SD algo-

---

[2]https://nvidia.github.io/TensorRT-LLM/advanced/speculative-decoding.html#ngram

rithms often relies on datasets like MT-Bench or domain-specific subsets for coding and math. For instance, *EAGLE3*, arguably the most widely adopted speculation method, validates performance on a combination of MT-Bench (limited to 10 samples per category with minimal intra-category variance), HumanEval (Chen et al., 2021) for coding (restricted to simple Python-only tasks), Alpaca (Taori et al., 2023) for instruction following (evaluated on the training set as no official test set exists), GSM8K (Cobbe et al., 2021) for math (confined to grade-school level tasks), and CNN/DM (See et al., 2017a). Crucially, because SD speedups are tied to the text domain, this methodology is not enough to represent the complexity of real-world data distribution. SpecBench represents the most significant step towards standardized evaluations. However, because it sources the majority of its data categories directly from MT-Bench, it inherits critical limitations regarding scale and diversity. Most categories contain as few as 10 samples limited to two turns. Additionally, most categories feature short mean ISLs ($< 100$ tokens) that may fail to stress modern drafters. Furthermore, SpecBench's supplemental categories often lack structural diversity. For example, the multilingual subset, sourced from WMT14 DE-EN (Bojar et al., 2014), consists entirely of translation prompts (*"Translate German to English:"*), and constitutes $\sim 15\%$ of the total dataset. We provide a detailed comparison of dataset statistics in Appendix A and an empirical analysis of these gaps in Section 8.3. Finally, current evaluation methodologies focus predominantly on $BS = 1$ in non-optimized environments, omitting throughput-oriented evaluation essential for real-world serving.

## 3. Background

**Speculative Decoding**  Standard LLM inference is autoregressive, meaning generating a sequence of length $T$ requires $T$ sequential forward passes. On modern GPUs, this process is *memory-bound* at low concurrencies: the latency is dominated by loading model parameters from HBM to on-chip caches, leaving compute units underutilized. SD addresses this by employing a lightweight draft model $M_d$ to predict $\gamma$ future tokens. The target model $M_t$ then verifies these candidates in a single parallel forward pass. Since verifying $\gamma$ tokens incurs comparable memory access costs to generating a single token, the verification overhead becomes relatively negligible at low concurrency. The system accepts the first $k$ tokens that match the target distribution, ensuring lossless acceleration via rejection sampling (Chen et al., 2023). The efficacy of SD is strictly tied to the serving regime. SD yields maximum speedups in memory-bound settings. As batch size increases, the system shifts toward a *compute-bound* regime, often diminishing the relative benefits of parallel verification, in some cases resulting in slowdowns, as exemplified in Section 8.1. We also note that Mixture-of-Experts (MoE) models are particularly well-suited for SD. While they have very large parameter counts, each token activates only a sparse subset of experts, so the system rarely becomes compute-bound. Verification requires loading additional experts for draft tokens, increasing memory access, but reaching a fully compute-bound state requires activating all experts across a massive number of tokens. As a result, SD can provide benefits for MoEs even at high batch sizes (Huang et al., 2025). This is demonstrated in Figure 7. The interplay between serving regimes and model architectures underscores the need for a benchmarking solution capable of quantifying these trade-offs.

**Metrics**  We utilize standard metrics for SD evaluation: conditional **Acceptance Rate (AR)** is defined as the probability of accepting draft token $x_i$ *given* the draft prefix $x_{<i}$ was accepted. **Acceptance Length (AL)** represents the expected number of generated tokens per verification step $L_t$ (including "free" verification token). For a draft length **(DL)** $\gamma$ and conditional acceptance rates $\text{AR}_i$, this is expressed as

$$\text{AL} = \mathbb{E}[L_t] = 1 + \sum_{i=1}^{\gamma} \prod_{j=1}^{i} \text{AR}_j. \tag{1}$$

Finally, we measure system efficiency via **Throughput (Output TPS)**, defined as the total tokens generated across all concurrent requests per second, and **User TPS**, defined as tokens generated per second for a single request. User TPS serves as a proxy for end-user latency.

## 4. Overview of SPEED-Bench

SPEED-Bench consists of a multi-purpose dataset and a unified measurement framework. The dataset is further divided into two splits, each tailored for specific constraints.

**Qualitative Split**  To measure speculation quality (ARs and ALs) it is critical to test across a wide range of semantic domains. Rather than performing an exhaustive and computationally expensive evaluation across dozens of data sources, we apply a specialized selection algorithm to 18 publicly available sources, to curate a compact subset that maximizes semantic diversity. This allows for relatively fast, high-fidelity measurements of how accurate a speculator is across fine-grained categories (e.g., Math, Coding, QA).

**Throughput Split**  Evaluating system speedups requires a different approach. Throughput measurements must account for batch size scaling and, ideally, precise ISL regimes that simulate diverse scenarios. To address this, the Throughput Split aggregates samples into three different difficulty categories, along fixed ISL buckets (1k-32k). This ensures sufficient data volume to construct stable throughput-latency Pareto curves, allowing the split to serve as a robust proxy for measuring system latency.

**Measurement Framework**  Complementing these data

splits is the measurement framework, compatible with both production-grade engines (SGLang, vLLM, TensorRT-LLM) and research-oriented libraries (SpecBench). By handling text processing externally, the framework ensures that all engines process identical sequences, isolating the impact of the speculation algorithm from the system implementation, thereby standardizing comparisons across engines.

## 5. SPEED-Bench Qualitative Split

The efficacy of a speculator is tied to the domain and entropy of the input text. For accurate measurements, the Qualitative Split is designed to maximize semantic coverage while minimizing the computational cost of evaluation. Rather than performing an exhaustive evaluation across disparate benchmarks, we construct a small but highly diverse subset of samples. In this section, we outline the data format and the selection algorithm used to curate this dataset.

**Data Composition**   We aggregate data from 18 publicly available datasets, splitting them into 11 distinct categories. We select 80 samples per category, resulting in a total of 880 samples. The selected categories were inspired by SpecBench, with a few refinements (details in Appendix A). The guiding principle for selecting data sources was semantic heterogeneity. While SpecBench offers a large number of categories, we found that the samples within them often lack internal diversity or are overly simplistic, hindering effective evaluation. For example, we found the *Coding* and *Humanities* categories, which are entirely based on MT-Bench, to be quite simple and not representative of the complexity found in modern LLM benchmarks. Furthermore, to facilitate fine-grained evaluation, we enrich the dataset with comprehensive metadata. Each sample includes a *subcategory* classification and a *multiturn* binary indicator; unlike SpecBench, which is limited to two turns, approximately 20% of our samples feature multi-turn interactions spanning two to five turns. We further provide a *difficulty* field for *Coding*, *Humanities*, *Math*, and *STEM*, focusing on hard problems (~80%) while retaining a subset of easier tasks for coverage. Finally, to ensure meaningful signal for speculation metrics, we verified that the samples generate a mean of ~650 tokens with GPT-4 (Achiam et al., 2023). See Appendix B for full details of the data gathering process.

**Selection Algorithm**   To keep the benchmark efficient yet representative, we employ a heuristic-based selection strategy designed to maximize the semantic diversity within each category. We map samples $t_i$ to dense vectors $x_i \in \mathbb{R}^d$ using a pre-trained text embedder, specifically OpenAI's *text-embedding-3-large* [3], and row-normalize such that $\|x_i\| = 1$, allowing cosine similarity to be computed as $x_i^\top x_j$. Given $N$ prompt candidates from various sources

---

[3]https://platform.openai.com/docs/guides/embeddings

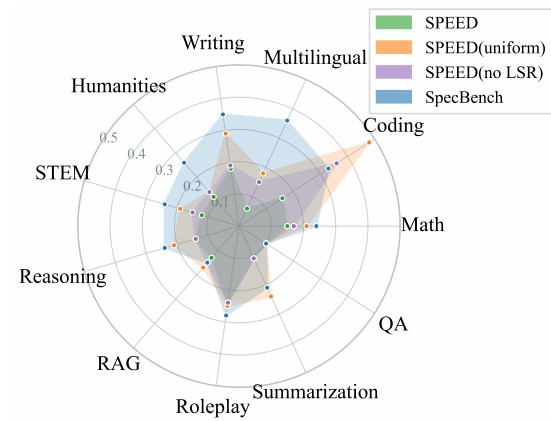

*Figure 2.* Comparison of average semantic similarity between samples **(lower is better)**. SPEED-Bench achieves lower similarity than both random selection and SpecBench across all categories. No LSR stands for no Local Swap Refinement.

and a target size $k$, we seek a subset $S \subset \{1...N\}, |S| = k$ that minimizes total pairwise similarity:

$$\mathcal{L}(S) = \sum_{i \in S} \sum_{j \in S, j \neq i} x_i^\top x_j \tag{2}$$

Minimizing this objective ensures that the selected samples span the semantic space as widely as possible, reducing redundancy. As finding an exact solution to this is NP-hard, we employ a **Greedy Selection Algorithm** with **Local Swap Refinement** (see Algorithm 1). We initialize $S$ with a random index and iteratively append $i^* = \mathrm{argmin}_{i \notin S} \sum_{j \in S} x_i^\top x_j$. To escape local minima, we then iteratively swap $i_{out} \in S$ with $i_{in} \notin S$ if the swap strictly decreases $\mathcal{L}(S)$, repeating until convergence. We additionally compare against a quadratic programming approximation. This and additional analysis on the selection algorithm are in Appendix C. Our selection algorithm reduces average semantic similarity by 40% compared to SpecBench (notably, 83% in *Multilingual*). As shown in Figure 2, our algorithm outperforms random selection on identical data, confirming the selection of diverse samples. Furthermore, the fact that random selection outperforms SpecBench on most categories, demonstrates the high quality of our selected data sources. See Appendix D for the pairwise similarity matrices of a few resulting subsets.

## 6. SPEED-Bench Throughput Split

The Throughput Split is tailored for evaluating system-level efficiency, addressing a gap in literature for benchmarking SD under high concurrency ($BS > 1$) and long ISLs. As concurrency increases, systems transition from memory-bound to compute-bound regimes, fundamentally altering the cost-benefit ratio of verification. SPEED-Bench addresses this by providing workloads to construct throughput-

---

**Algorithm 1** Greedy Selection with Local Swap Refinement

---

**Require:** Candidates $X \in \mathbb{R}^{N \times d}$ (row-normalized), target size $k$
**Ensure:** $S \subset \{1, \ldots, N\}$
1: $S \leftarrow \{i_{rand}\}, m \leftarrow Xx_i$
2: **while** $|S| < k$ **do**
3:   $i^* \leftarrow \arg\min_{j \notin S} m_j$;  $S \leftarrow S \cup \{i^*\}$;  $m \leftarrow m + Xx_{i^*}$
4: **end while**
5: **repeat**
6:   Find swap pair $(i_o \in S, i_i \notin S)$ that minimizes $\Delta$:
7:     $\Delta = \sum_{j \in S \setminus \{i_o\}} (x_{i_i}^\top x_j - x_{i_o}^\top x_j)$
8:   **if** $\Delta < 0$ **then**
9:     $S \leftarrow (S \setminus \{i_o\}) \cup \{i_i\}$
10:   **end if**
11: **until** convergence or max_iter
12: **Return** $S$

---

latency Pareto curves across diverse serving scenarios.

**The Pitfalls of Synthetic Benchmarking**  A common practice in inference benchmarking is the use of random token batches to simulate prompt load. While effective for measuring autoregressive decoding in dense models, it is fundamentally flawed for evaluating SD, where performance depends on the predictability of the input distribution. Random tokens trigger two primary failure modes that skew AR measurements: **1) Trivial Response:** The model identifies noise and defaults to predictable acknowledgments, artificially inflating ARs. **2) Topic Latching:** The model anchors to keywords within the noise, hallucinating a coherent but arbitrary response, typically resulting in lower ARs. Examples are in Appendix E. In Section 8.4 we empirically demonstrate the differences in throughput measurements.

Furthermore, we find that random noise fails to trigger realistic expert routing in MoE architectures (see Appendix G). Routers may "collapse" to a subset of experts, violating load-balancing assumptions and causing inaccurate step latency measurements even without SD. SPEED-Bench utilizes real data to ensure measurements translate to real environments.

**Data Composition**  To isolate the effects of context length, we construct fixed ISL buckets (1k, 2k, 8k, 16k, 32k), sourcing data from 8 publicly available datasets. We ensure uniformity by either truncation where possible, or by padding prompts with a neutral suffix. ISLs are calculated using the `o200k_base` tokenizer. This ensures deterministic load during prefill while preserving the prompt's semantics. Samples are aggregated into three broad categories based on domain entropy—**Low Entropy** (e.g., sorting and coding), **Mixed Entropy** (e.g., STEM), and **High Entropy** (e.g., creative writing), following the taxonomy in Li et al. (2025a). Each of the five ISL buckets contains 512 samples per category (1,536 total per bucket), enabling the construction of stable throughput-latency Pareto curves. Unlike the Qualitative Split, we do not filter for fine-grained domains here, as replicating such granularity at this scale with high

quality is both impractical and as discussed next, perhaps redundant for speedup estimations. We verify that the samples generate a mean of $\sim 2.4k$ tokens (on GPT-4 with 16k ISLs). See Appendix B for full details on data gathering.

**Estimating Domain-Specific Speedups**  Another capability of the Throughput Split is that it allows practitioners to estimate speedups for fine-grained categories under specific batch and ISL constraints, without the need to gather the high volume of valid data required for stable measurements. While AL is domain-dependent, we note that *per-step latency* is primarily governed by system constraints (e.g., memory bandwidth) and serving parameters (e.g., batch size, ISL). Given reliable measurements for the per-step latency of standard autoregressive decoding ($t_{ar}$) and SD ($t_{sd}$), speedup can be analytically inferred as Speedup $= (t_{ar} \cdot AL)/t_{sd}$. Extended details are in Appendix H. For this proxy to be accurate, the latency measurements must be realistic. As discussed before, random tokens fail to provide reliable estimates for various reasons. However, the Throughput Split provides the realistic workloads necessary for reliable measurements of $t_{ar}, t_{sd}$.

## 7. Measurement Framework

A critical challenge in benchmarking SD across inference engines is the variability in how they handle and process raw text inputs. For instance, different engines may inconsistently append *Beginning of Sequence* (BOS) tokens or apply chat templates, thereby changing the drafted sequence and complicating cross-engine evaluation. To mitigate this, we introduce a unified and lightweight evaluation framework, that operates as a thin client and ensures that the draft and target models across all backends process identical token sequences, isolating the performance impact of the speculation algorithm and the engine's system optimizations. We draw a distinction between *external factors* and *internal factors*. External factors, such as tokenization, prompt templating, and special token handling, are normalized to eliminate evaluation noise and guarantee apples-to-apples comparisons of the SD algorithms. In contrast, internal factors, including numerical implementations, CUDA kernel optimizations, scheduler design, and continuous batching mechanisms, are fundamental characteristics of the serving framework. Our goal is not to abstract away engine-specific optimizations, but rather to measure how SD performs end-to-end within realistic deployment environments.

**Metrics and Concurrency**  The framework operates on an asynchronous event loop powered by Python's *asyncio*, enabling the concurrent dispatch of requests to mimic high-throughput serving scenarios. We capture fine-grained performance data by analyzing the streaming response objects returned by the inference engine. ARs are calculated by inspecting the number of newly generated tokens per response

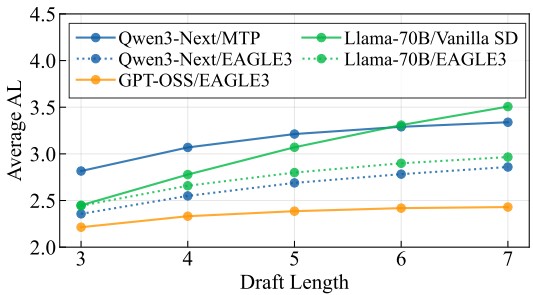

*Figure 3.* Average AL on the Qualitative Split. External drafting scales better across DLs.

chunk; a chunk containing multiple tokens indicates a successful speculation step. Timestamps are recorded upon the receipt of every object to compute latency metrics, including Time To First Token (TTFT), step latency, and total request latency. These allow us to derive aggregate metrics such as User TPS and overall output TPS (throughput). While SD is fundamentally a decode-phase optimization, the TTFT metric is reported for completeness. While *asyncio* provides sufficient concurrency for most batch sizes, we note that at extremely high throughputs ($BS > 256$), the Python Global Interpreter Lock (GIL) can introduce client-side overheads. We highlight this as a **current limitation**, and are extending the framework to leverage more advanced parallelism to ensure fidelity in these regimes. Note that in batched SD, different sequences may accept different numbers of tokens per verification step. Modern inference engines handle this divergence through KV-cache rollback while maintaining aligned batch shapes during target verification.

**Ecosystem Integration** Native integration is provided for industry-standard frameworks including SGLang, vLLM, and TensorRT-LLM, allowing users to evaluate SD performance while leveraging optimizations such as CUDA Graphs, continuous batching, and kernel fusion. We emphasize that SPEED-Bench is designed to *complement*, not replace, existing research toolkits like SpecBench. While the SpecBench framework excels at evaluating methods using native PyTorch/HuggingFace, SPEED-Bench focuses on the viability of these methods in deployment. To support a holistic pipeline, we demonstrate how SpecBench models can be evaluated within our framework. The supplementary material includes an example for SpecBench's Medusa, and instructions for further model extensions.

## 8. Experiments and Observations

In this section we demonstrate the versatility of SPEED-Bench and build intuition regarding SD performance in real-world scenarios. We focus our evaluation on SD methods integrated into production-grade frameworks, including N-Grams, Vanilla SD (external drafting), EAGLE3, and native

MTP heads. Our experiments target large, modern open models: Llama 3.3 70B, GPT-OSS 120B, Qwen3 235B, Qwen3-Next, and DeepSeek R1. We exclusively utilize draft chains rather than tree-based verification. While the measurement framework technically supports tree-based evaluation, we focus on draft chains as they remain the standard for $BS > 1$ speedups in production engines (Li et al., 2025b). Appendix F provides additional experiments with tree-based verification.

For Vanilla SD, we use the following draft-target pairs: Llama 3.2 1B with Llama 3.3 70B and Qwen3 0.6B with Qwen3 235B. For EAGLE3, we utilize **pretrained open checkpoints**, except where custom training is mentioned. MTP is used on models with native support. All experiments used a single NVIDIA B200 GPU, except for DeepSeek and Qwen models inference and GPT-OSS EAGLE3 training, which used eight. Additional details are in Appendix I.

### 8.1. Measuring Speculator Accuracy and Speedups

We evaluate speculation accuracy and system speedups across the Qualitative Split. All measurements use a batch size of 32 to simulate realistic workloads, utilizing TensorRT-LLM and SGLang for Qwen3 models due to engine constraints. Table 1 presents the average ALs and speedups using a DL of 3. The results confirm a correlation between domain entropy and performance: low-entropy domains, such as *Coding* and *Math*, yield larger ALs than high-entropy tasks like *Roleplay*. Notably, we observe net slowdowns with N-Gram speculation at this concurrency, as ARs fail to justify validation costs. Furthermore, in some settings, Vanilla SD yields lower speedups than EAGLE3 despite comparable ALs, due to external model overhead.

Figure 3 illustrates AL scaling across DLs, highlighting two trends: **1) Native MTP Robustness:** Qwen3-Next's pre-trained MTP head maintains higher ALs than the post-trained setups like EAGLE3. This suggests that pretraining offers significant accuracy gains. **2) Vanilla SD Scaling:** Despite higher draft overhead, external drafting sustains accuracy better than EAGLE3 at longer speculation horizons.

**Measuring ALs on the Throughput Split** We analyze AL stability across the ISL buckets of the Throughput Split. As expected, we observe an inverse correlation between category complexity and ALs regardless of context length: *High Entropy* prompts yield the lowest ALs, while *Low Entropy* prompts yield higher ALs. While general trends hold, specific deviations attributed to training data distributions exist. Due to space constraints, results are in Appendix J.

### 8.2. Vocabulary Pruning Effects

Limited semantic diversity can obscure the negative side effects of aggressive optimizations. As an example, EA-

*Table 1.* Average AL and speedups on the Qualitative Split, measured using $BS = 32$ and a DL of 3. (–) indicates a method that is not currently supported in the running framework. (*) indicates measurements on SGLang.

| Domain | Llama 3.3 70B | | | GPT-OSS 120B | | DeepSeek R1 | Qwen3 235B (*) | | Qwen3-Next (*) | |
|---|---|---|---|---|---|---|---|---|---|---|
| | N-Gram | Vanilla | EAGLE3 | N-Gram | EAGLE3 | MTP | Vanilla | EAGLE3 | EAGLE3 | MTP |
| | | | | | | Temperature=0 | | | | |
| Coding | 1.54 | 2.72 | 3.00 | 1.31 | 2.46 | 2.76 | 2.62 | 2.26 | 3.17 | 3.34 |
| Humanities | 1.39 | 2.30 | 2.34 | 1.35 | 2.28 | 2.53 | 2.32 | 2.13 | 2.22 | 2.68 |
| Math | 1.43 | 2.43 | 2.45 | 1.30 | 2.46 | 2.77 | 2.69 | 2.37 | 2.90 | 3.13 |
| Multilingual | 1.91 | 2.59 | 1.71 | 1.58 | 2.32 | 2.68 | 2.87 | 2.33 | 1.90 | 3.19 |
| QA | 1.21 | 2.35 | 2.35 | 1.27 | 2.25 | 2.52 | 2.28 | 2.23 | 2.13 | 2.71 |
| RAG | 1.51 | 2.50 | 2.76 | 1.31 | 2.31 | 2.61 | 2.54 | 2.44 | 2.46 | 2.94 |
| Reasoning | 1.34 | 2.46 | 2.61 | 1.31 | 2.39 | 2.62 | 2.51 | 2.32 | 2.60 | 2.89 |
| Roleplay | 1.15 | 2.14 | 2.04 | 1.25 | 1.87 | 2.14 | 1.92 | 1.87 | 1.80 | 2.09 |
| STEM | 1.38 | 2.45 | 2.39 | 1.30 | 2.28 | 2.62 | 2.44 | 2.18 | 2.47 | 2.85 |
| Summarization | 1.36 | 2.47 | 2.59 | 1.25 | 2.13 | 2.47 | 2.37 | 2.26 | 2.19 | 2.66 |
| Writing | 1.33 | 2.45 | 2.63 | 1.20 | 1.98 | 2.33 | 2.19 | 2.02 | 2.09 | 2.46 |
| **Mean AL** | 1.41 | **2.44** | **2.44** | 1.31 | **2.25** | 2.55 | **2.43** | 2.22 | 2.36 | **2.81** |
| **Mean Speedup** | 0.88× | 1.60× | **1.90×** | 0.29× | **1.34×** | 1.45× | 1.17× | **1.33×** | 1.06× | **1.20×** |
| | | | | | | Temperature=1 | | | | |
| **Mean AL** | 1.37 | 1.98 | 2.37 | 1.27 | 2.07 | (–) | 2.21 | 2.03 | 2.26 | 2.68 |
| **Mean Speedup** | 0.85× | 1.15× | 1.75× | 0.27× | 1.06× | (–) | 1.06× | 1.21× | 1.03× | 1.18× |

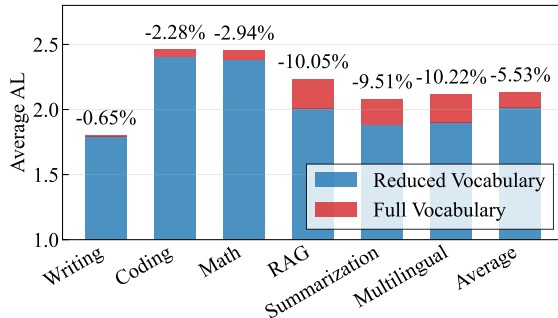

*Figure 4.* Average AL across selected categories using GPT-OSS 120B and EAGLE3 drafters (full vs. pruned vocabulary), $DL = 3$.

GLE3 applies *vocabulary pruning* (usually to 32k tokens), mitigating the computational bottlenecks of the final projection layer. While effective for standard inputs, this heuristic degrades performance on the "long tail" of user inputs. We identified the *Multilingual* category as a particular weakness, where approximately 22% of target tokens are missing from the pruned vocabulary (Appendix K). As shown in Figure 4, our empirical results demonstrate high variance in how different domains respond to pruning. While the performance drops are relatively negligible in *Math* and *Coding*, and results remain comparable in *Writing*, we observe significant accuracy degradation in other domains. Specifically, the largest impacts are observed in the *Multilingual*, *RAG*, and *Summarization* categories. These results underscore the need for broad-coverage evaluation to ensure that draft latency does not come at the cost of generalization.

## 8.3. Comparison with SpecBench

While SpecBench provided a foundational step toward standardized evaluation, its limited volume and semantic scope can lead to misleading conclusions regarding algorithmic efficacy. In SpecBench, categories such as *Coding* and *Reasoning* contain only 10 samples each, creating statistical noise where the lightweight EAGLE3 drafter appears comparable to more robust Vanilla SD methods, as shown in Figure 5. SPEED-Bench corrects this impression: by evaluating on larger, semantically diverse splits, we reveal the expected advantage of the external drafter at long DLs. This necessity for diversity is even more pronounced in the *Multilingual* category. SpecBench's multilingual subset consists entirely of German-to-English translation prompts, whereas SPEED-Bench encompasses a broad spectrum of languages and tasks. Consequently, while SpecBench shows only a moderate performance gap, SPEED-Bench reveals a substantial advantage for the external drafter. Notably, the *Multilingual* and *Coding* categories are where our selection algorithm achieved the highest reduction in semantic similarity (Figure 2), confirming that high-diversity benchmarks are essential to expose differences between methods.

## 8.4. Measuring Latency and Throughput

In this section, we demonstrate the utility of SPEED-Bench for evaluating system efficiency in realistic serving scenarios. Unlike methods that focus on latency at $BS = 1$, SPEED-Bench enables the construction of throughput-latency Pareto curves, providing insights into the interplay

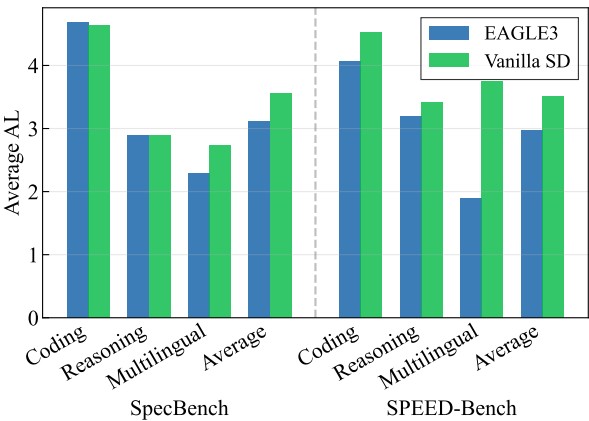

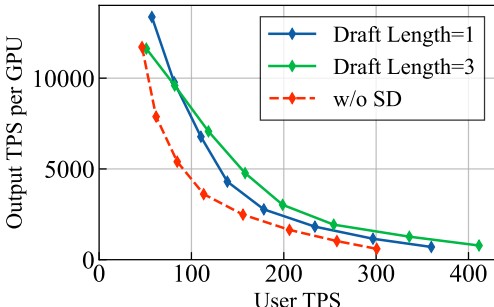

*Figure 7.* Throughput as a function of user TPS, comparing $DL = 1, 3$ on the Throughput Split (2k). Target is GPT-OSS 120B with EAGLE3, measured on vLLM. Points represent BS from 2 to 256.

*Figure 5.* Average AL across selected categories in SpecBench vs SPEED-Bench. Target model is Llama 3.3 70B. $DL = 7$. Full results are in Appendix L.

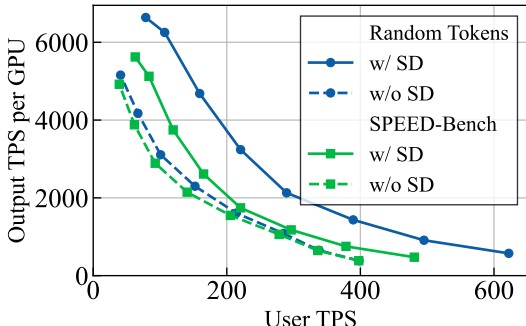

*Figure 6.* Throughput as a function of user TPS, comparing random input tokens to the Throughput Split (8k). Target is GPT-OSS 120B with EAGLE3 drafter, measured on TensorRT-LLM. $DL = 3$. Points represent BS from 1 to 128.

between BS, DL, and inference engines.

**Random data VS SPEED-Bench** In Section 6, we identified the risks of benchmarking with random token inputs. Figure 6 confirms this empirically. With SD enabled, synthetic benchmarking overestimates throughput by an average of 23% compared to SPEED-Bench, confirming the impact of skewed ARs. Interestingly, a performance gap is also observable in the baseline autoregressive setting. We attribute this discrepancy to the expert imbalance described in Appendix G: random inputs fail to trigger realistic expert routing in the MoE target model. This leads to inaccurate step latency measurements even without speculation.

**Optimal DL selection** Figure 7 illustrates the impact of DL on system throughput across varying batch sizes. We observe that the optimal DL shifts depending on the concurrency regime. At lower batch sizes, where the system is memory-bound, longer drafts are preferred. However, as the batch size increases and the system approaches the compute-bound regime, the cost of verifying additional tokens may

outweigh the gains, and $DL = 1$ is preferred. SPEED-Bench allows practitioners to identify these crossover points and select the optimal DL for their constraints.

**Impact of Inference Framework** We also investigate how different inference engines affect speedups. In summary, we find that TensorRT-LLM achieves higher peak throughput by leveraging a unified CUDA graph for the entire draft-verification loop. In contrast, vLLM's multi-engine design incurs slight host communication overheads, though we note it offers greater flexibility for dynamic drafting strategies. Due to space constraints, this analysis is in Appendix M.

### 8.5. Training Data ISL Effects

The Throughput Split allows us to test the stability of drafters at long ISLs. During our experimentation, we identified two publicly available EAGLE3 models for GPT-OSS 120B, and found that both suffer from significant degradation at high ISLs (see Appendix N). While the exact cause is unconfirmed, these drops may be attributed to either insufficient training ISLs or missing RoPE scaling configurations.

To quantify the effects of training ISLs and RoPE scaling, we train EAGLE3 models for GPT-OSS 120B at varying sequence lengths (1k, 2k, 4k) and evaluate them across SPEED-Bench's ISL buckets. Training configuration is in Appendix I. As expected, Figure 8 shows that accuracy degrades rapidly when the inference ISL exceeds the training ISL. Notably, we were easily able to evaluate potential mitatigations using SPEED-Bench. For example, we show that simply applying YaRN scaling (Peng et al., 2024) at inference time recovers significant drafting accuracy, even for models trained on relatively short sequences ($\geq$ 2k).

## 9. Conclusion

SPEED-Bench establishes a unified evaluation ecosystem for both SD research and production-grade deployment. By providing a semantically diverse Qualitative Split and a

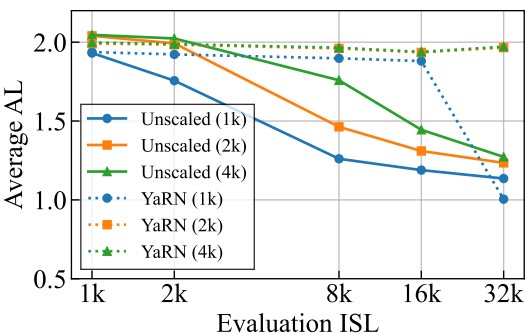

*Figure 8.* Average AL as a function of ISLs, comparing training ISLs. Target is GPT-OSS 120B with EAGLE3 drafters, measured on vLLM. Dotted lines denote YaRN scaling. Legend labels (1k, 2k, 4k) indicate the maximum ISL used during training. $DL = 3$.

Throughput Split focused on large batches and fixed ISLs, the framework enables the analysis of critical system properties. Specifically, these splits allow practitioners to quantify the robustness of their methods across diverse text domains, identify batch-size and ISL dependent optimal DLs, and measure speedups across varied serving settings. Our empirical results demonstrate that SD performance is deeply data-dependent and sensitive to the specific serving regime. We hope SPEED-Bench facilitates a shift towards more rigorous evaluation standards, enabling the community to develop SD methods that remain robust and efficient when deployed in production environments.

## Impact Statement

This paper presents work whose goal is to advance the field of Machine Learning. There are many potential societal consequences of our work, none which we feel must be specifically highlighted here.

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

## A. Detailed Data Comparison with SpecBench

Table 2 compares SPEED-Bench with SpecBench. SPEED-Bench's Qualitative Split is inspired by SpecBench and covers 11 categories: *Coding*, *Math*, *Humanities*, *STEM*, *Writing*, *Summarization*, *Roleplay*, *RAG*, *Multilingual*, *Reasoning*, and *QA*. To avoid redundancy, the SpecBench categories *Math* and *Math Reasoning* are consolidated into *Math*, and *Extraction* and *RAG* into a single *RAG* category. We further replace the *Translation* category with *Multilingual*, incorporating a broader set of languages and additional multilingual tasks. Samples are uniformly distributed across all 11 categories. SPEED-Bench sources data from 24 distinct datasets, compared to 5 in SpecBench. The use of multiple sources enables broader task coverage; for example, coding tasks span multiple programming languages and question types, while multilingual tasks extend beyond simple translation to include diverse queries in different languages. Table 3 compares the per-category average ISL in the Qualitative Split compared to SpecBench. Note that the median ISL in SpecBench is $\sim 57$ tokens per sample, whereas SPEED-Bench more than doubles this to $\sim 141$ tokens.

In addition, SPEED-Bench features longer multi-turns samples and introduces new metadata fields, including task difficulty and sub-categories, enabling fine-grained analysis of SD algorithms. Difficulty labels are assigned for the *Coding*, *Math*, *Humanities*, and *STEM* categories based on the upstream benchmark definitions. For *Math*, *STEM*, and *Humanities*, MT-Bench samples are labeled as *medium*, Humanity's Last Exam samples as *hard*, and GSM8K samples as *easy*. For *Coding*, MT-Bench samples are similarly labeled as *medium*. LiveCodeBench samples retain the original difficulty labels provided by the benchmark, while CodeContests difficulties are normalized into a unified schema by mapping *hard*, *harder*, and *hardest* to *hard*, and converting levels *A–I* to *easy*, *J–Q* to *medium*, and *R–Z* to *hard*.

SPEED-Bench also introduces the Throughput Split, enabling measurements of system and user TPS in large-batch fixed-ISL (1k-32k) regimes. A similar data split is not available in SpecBench.

*Table 2.* Comparison between SPEED-Bench and SpecBench across different metrics.

|  | **SPEED-Bench** | **SpecBench (Xia et al., 2024)** |
|---|---|---|
| # Samples per Category | 80 (qualitative), $512 \times 3$ (throughput) | 10 (for 8 categories), 80 (the rest) |
| # Total Samples | 880 (qualitative), $1536 \times 5$ (throughput) | 480 |
| # Data Sources | 24 | 5 |
| Avg. Pairwise Similarity (Figure 2) | 0.14 | 0.22 |
| # Multiturn Prompts | 167 | 80 |
| Max # Turns | 5 | 2 |
| Subcategories | ✓ | ✗ |
| Difficulties | ✓ (for *Math*, *STEM*, *Humanities*, *Coding*) | ✗ |
| Long ISLs (16k-32k) | ✓ | ✗ |
| Large batches of fixed-size ISLs | ✓ | ✗ |
| Programming Lanuages Explicitly Mentioned in *Coding* | Python (27), CPP (9), Java (10), Go (13), Javascript (11), Rust (3), HTML (1), CSS (1) | Python (3), CPP (1), HTML (1), CSS (1) |
| # Distinct Lanuages in *Multilingual* Lanuages in *Multilingual* | 23 EN, DE, ZH, IT, MG, FR, JA, PT, AR, MK, DA, NL, KO, ES, NN, TH, VI, BN, GU, CS, GD, EU, RU | 2 EN, DE |
| Difficulty level in *Math*, *Humanities* and *STEM* categories | Academic level | High school level |

*Table 3.* Comparison of mean ISL by category between SpecBench and SPEED-Bench datasets. Parenthesis indicate median ISL.

| Category | SpecBench | SPEED-Bench |
|---|---|---|
| *Coding* | 72.5 (39.0) | 218.4 (146.5) |
| *humanities* | 56.1 (36.5) | 99.3 (51.0) |
| *Math* | 58.5 (45.0) | 130.5 (98.5) |
| *QA* | 11.1 (11.0) | 11.1 (11.0) |
| *RAG* | 683.6 (682.0) | 805.7 (667.0) |
| *Reasoning* | 86.3 (64.0) | 163.8 (88.5) |
| *Roleplay* | 74.3 (74.0) | 464.4 (215.0) |
| *STEM* | 54.3 (55.0) | 114.5 (74.0) |
| *Summarization* | 710.4 (646.0) | 581.1 (387.5) |
| *Multilingual* | 34.8 (32.0) | 160.4 (130.5) |
| *Writing* | 59.9 (54.0) | 619.3 (337.5) |
| Macro-average | 169 | 406 |

# B. Extended Details on Dataset Collection

This appendix provides detailed information about the data sources and construction methods used for each component of SPEED-Bench. Table 4 summarizes data sources for the Qualitative Split, and Table 5 summarizes data sources for the Throughput Split. For the Throughput Split, we ensure fixed ISLs by either truncation where possible, or by padding prompts with the neutral suffix *"please answer now"*.

*Table 4.* Data sources and construction methods for the Qualitative Split.

| Source | Categories | Construction Details |
|---|---|---|
| SpecBench | All besides Summarization and Multilingual | Used directly from source. |
| Humanity's Last Exam (Phan et al., 2025) | STEM, Humanities, Math | Filtered to text-only samples (no images) with exact-match answer type. For STEM: filtered to Physics, CS/AI, Biology/Medicine, Chemistry, Engineering. For Humanities: filtered to Humanities/Social Science category. |
| LiveCodeBench Lite (Jain et al., 2024) | Coding | Constructed instruction prompts requesting code generation in a randomly selected programming language (Python, Java, C++, Go, JavaScript, Rust). Includes starter code when available. |
| Code Contests (Li et al., 2022) | Coding | Constructed instruction prompts requesting program generation in a randomly selected language (Python, Java, C++). Problem descriptions used directly from source. |
| HumanEvalPack (Muennighoff et al., 2023) | Coding | Used code completion prompts directly from source. |
| RoleBench (Wang et al., 2023) | Roleplay | Constructed multi-turn roleplay prompts using role descriptions and questions. Questions grouped by role into conversations (1–5 turns). System prompts randomly sampled from 8 prompt templates instructing the model to embody the character. |
| CoSER (Wang et al., 2025) | Roleplay | Constructed roleplay prompts with character profiles, scenario, and character motivation, only for books that are available in the public domain. |
| WritingBench (Wu et al., 2025) | Writing | Filtered to English samples. Writing queries used directly as single-turn prompts. |
| Creative Writing V3 (Paech, 2025) | Writing | Expanded prompts by replacing <SEED> placeholders with the seed modifiers provided, creating multiple variations per base prompt. |
| MT-Bench 101 (Bai et al., 2024) | Reasoning | Filtered to general reasoning and mathematical reasoning tasks. |
| MMLU-Pro (Wang et al., 2024b) | Reasoning | Grouped questions by category and combined multiple questions together to create multi-turn samples. |
| MMATH (Luo et al., 2025) | Multilingual | Questions used directly from source. |
| OPUS-100 (Zhang et al., 2020) | Multilingual | Constructed translation prompts by prepending "Translate the following text from [source language] to [target language]:". |
| MCIF (Papi et al., 2025) | Multilingual | Selected prompts for QA, translation, and summarization tasks with *long_mixed-prompt* format. |
| ChatRAG-Bench (Liu et al., 2024b) | RAG | Constructed prompts with context (concatenated retrieved passages) and multi-turn questions for the *hybridial* and *sqa* splits. |
| MCIF (Papi et al., 2025) | RAG | Used English QA prompts with *long_mixed-prompt* format, grouping questions by document into multi-turn conversations. |
| CNN/DailyMail (See et al., 2017b) | Summarization | Used articles directly from source with instructions to summarize the content. |

**Licensing and Filtering** For datasets involving code from GitHub repositories (e.g., Long Code Arena, RepoBench), we filter to include only repositories with permissive licenses (MIT License or Apache License 2.0) to ensure compliance with open-source licensing requirements. For roleplay data from CoSER, we verify that source books are in the public domain by checking their copyright status on Project Gutenberg, excluding any copyrighted works. For all other datasets, we ensure that our collection mechanisms adhere to the usage requirements specified by the data providers, including licensing, terms of service, and any other applicable guidelines.

*Table 5.* Data sources and construction methods for the Throughput Split.

| Source | Entropy Category | Construction Details |
|---|---|---|
| BAMBOO (Dong et al., 2023) | High entropy | Used MeetingPred and ShowsPred subsets. Constructed dialogue completion prompts asking the model to continue conversations. For longer contexts ($> 16k$ tokens), concatenated multiple dialogues. Padded/truncated to target token count. |
| Project Gutenberg (Project Gutenberg) | High entropy | Constructed book continuation prompts. Filtered to books with sufficient length and padded/truncated to target token count. |
| WritingBench (Wu et al., 2025) | High entropy | Reused English writing prompts from Qualitative Split. Filtered to prompts within 0.7–2$\times$ target token count, then padded/truncated. |
| AdaLEval (StackSelect) (Wang et al., 2024a) | Mixed | Constructed needle-in-a-haystack prompts asking models to select the most helpful answer from a set of StackOverflow answers and provide explanations for each choice. Padded/truncated to target token count. |
| Humanity's Last Exam (Phan et al., 2025) | Mixed | Used 50% of HLE data for few-shot prompting. Constructed prompts with category-specific demonstrations sampled from held-out examples, followed by the target question. Padded/truncated to target token count. |
| Long Code Arena (Bogomolov et al., 2024) | Low entropy | Used project-level code completion subset. Constructed prompts with repository context and file with `[COMPLETE]` markers for line-level completion. |
| RepoBench Python (Liu et al., 2024a) | Low entropy | Constructed cross-file code completion prompts with repository context snippets and in-file code. Padded/truncated to target token count. |
| RepoBench Java (Liu et al., 2024a) | Low entropy | Same construction as RepoBench Python but for Java code. |
| AdaLEval (TextSort) (Wang et al., 2024a) | Low entropy | Modified original sorting task to require outputting sorted text segments in order rather than just returning indices. Padded/truncated to target token count. |

# C. Alternative Selection Algorithm for the Qualitative Split

In this appendix, we analyze alternative benchmark construction strategies. Specifically, we compare against uniform random sampling and a convex quadratic programming (QP) approximation of the diversity optimization objective.

## C.1. Comparison Against Uniform Sampling

To evaluate whether the proposed selection algorithm meaningfully improves benchmark stability beyond simple random sampling, we compare it against uniform benchmark selection. We generate 10 independent uniformly sampled benchmarks and 10 benchmarks constructed using our optimization procedure (with different random seeds), all drawn from the same source pools while preserving the same number of samples per category. Each subset is evaluated using Llama 3.3 70B as target model with EAGLE3 as draft model.

**Stability of Per-Category Measurements**    A key requirement for a reliable benchmark is that category-level measurements remain stable across different benchmarks realizations. Under uniform sampling, we observe substantial variance in average AL measurements across benchmarks, particularly for high-variance domains such as Roleplay and Multilingual tasks. In contrast, our optimized selection procedure significantly reduces this instability. Table 6 reports the relative range of AL measurements across subsets, defined as the difference between the maximal AL and the minimal AL, normalized by the average AL.

*Table 6.* Relative variation of AL across independently generated benchmarks at $DL = 7$. Lower values indicate greater stability.

| Category | Uniform | Ours |
|---|---|---|
| Roleplay | 22.7% | **0.6%** |
| Multilingual | 14.6% | **5.1%** |
| RAG | 12.1% | **0.9%** |
| Humanities | 9.2% | **3.0%** |
| Writing | 7.2% | **0.9%** |
| STEM | 6.5% | **2.8%** |
| Math | 5.6% | **1.6%** |
| Summarization | 4.4% | **2.0%** |
| Coding | 4.0% | **0.7%** |
| Reasoning | 3.9% | **2.7%** |
| QA | **2.2%** | 2.6% |

**Consistency of Category Rankings**    Beyond absolute AL values, a benchmark should also preserve consistent relative ordering between categories. In practice, conclusions regarding which domains are "easy" or "hard" for SD should not depend on the specific random benchmark sampled. To quantify ranking consistency, we compute the normalized Kendall-$\tau$ distance between category rankings induced by different benchmarks, measuring the fraction of pairwise category orderings that disagree. As shown in Table 7, our optimized selection algorithm substantially improves ranking consistency. At $DL = 7$, the mean Kendall-$\tau$ disagreement is reduced by approximately $14\times$ relative to uniform sampling. Notably, 8 out of 9 optimized benchmarks produce an identical category ranking, whereas only 2 out of 10 uniformly sampled benchmarks share the same ranking, resulting in 8 distinct rankings among the uniform benchmarks.

*Table 7.* Normalized Kendall-$\tau$ disagreement between category rankings induced by independently generated benchmarks. Lower values indicate more consistent benchmark conclusions across subset realizations.

| Algorithm | $DL$ | Mean K-$\tau$ | Max K-$\tau$ |
|---|---|---|---|
| Uniform Sampling | 3 | 0.046 | 0.091 |
| Ours | 3 | **0.019** | **0.055** |
| Uniform Sampling | 7 | 0.056 | 0.127 |
| Ours | 7 | **0.004** | **0.018** |

## C.2. Quadratic Programming Approximation

Beyond the greedy selection algorithm detailed in the paper, we also explored a convex relaxation of the problem, formulating it as a quadratic programming (QP) task where we solve for a weight vector $w \in [0,1]^N$ minimizing $w^\top G w$ (where $G = XX^\top$ is the Gram matrix) subject to $\sum w_i = k$.

While the QP approach also yields high-quality solutions, our empirical tests showed that the Greedy Selection combined with Swap Refinement achieved similar diversity scores while being faster and more scalable to large candidate pools. Table 8 presents results for a few selected categories.

*Table 8.* Average pairwise similarity between samples in subsets constructed using different methods. Lower scores indicate better semantic diversity.

| Category | Random Selection | Greedy + Swap | QP Approximation |
|---|---|---|---|
| Writing | 0.29 | 0.18 | 0.18 |
| Humanities | 0.14 | 0.12 | 0.11 |
| RAG | 0.17 | 0.13 | 0.13 |
| Roleplay | 0.25 | 0.24 | 0.24 |

# D. Visualizing Semantic Diversity

Here we provide a qualitative comparison of the internal diversity between SPEED-Bench and SpecBench. Figure 9 and Figure 10 display the pairwise cosine similarity matrices for two categories: *Translation/Multilingual* and *Math*, respectively.

In these heatmaps, darker green values indicate high semantic similarity (redundancy), while lighter yellow values indicate low similarity (diversity).

- **SpecBench (Left Column):** This figure reveals clusters of highly repetitive prompts (e.g., the same math problem with minor changes, or identical translation structures), which might skew evaluation by over-weighting specific domains.

- **SPEED-Bench (Right Column):** Displays a pattern of low similarity. The absence of dark blocks confirms that our Greedy Selection Algorithm with Local Swap Refinement effectively minimizes redundancy, ensuring that the selected samples are semantically distinct and provide broader coverage of the task domain.

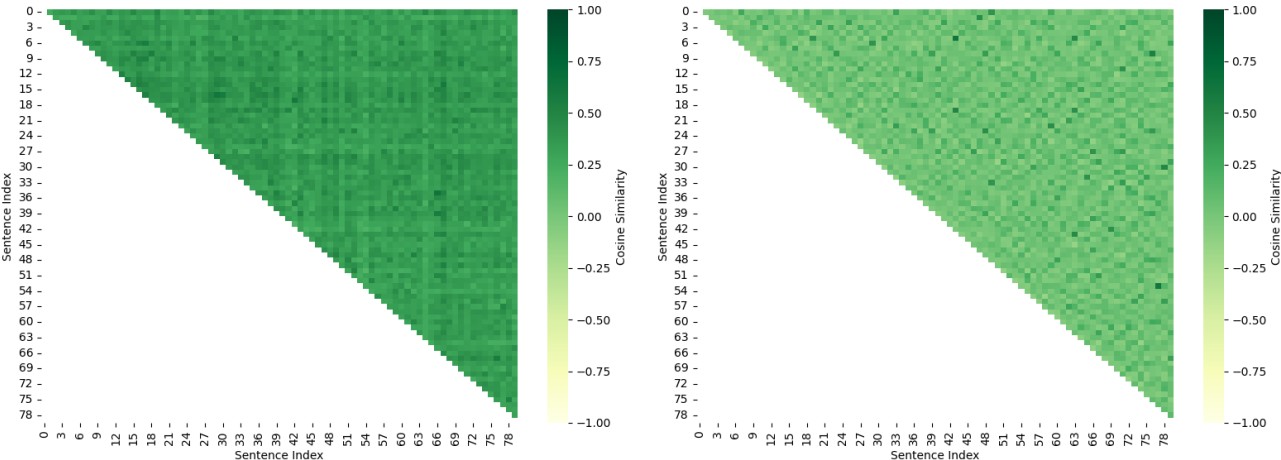

*Figure 9.* Pairwise similarity matrices for the 'Translation/Multilingual' category. SpecBench (left) shows dense blocks of high similarity, indicating redundant data. SPEED-Bench (right) shows a dispersed, low-similarity distribution, demonstrating better semantic diversity.

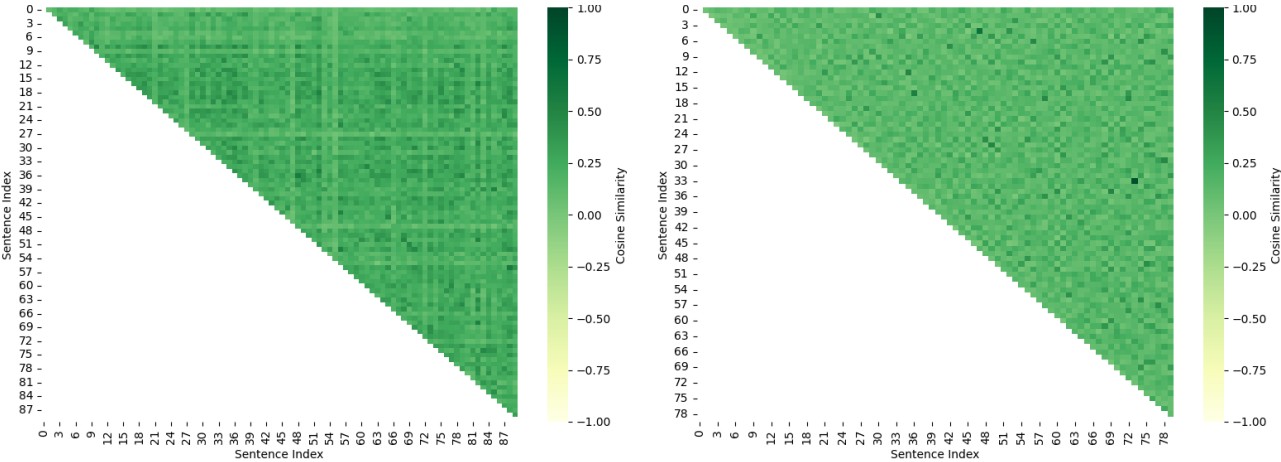

*Figure 10.* Pairwise similarity matrices for the 'Math' category. SpecBench (left) shows dense blocks of high similarity, indicating redundant data. SPEED-Bench (right) shows a dispersed, low-similarity distribution, demonstrating better semantic diversity.

# E. Pitfalls of Synthetic Benchmarking

In Section 6, we discuss why using random tokens for SD benchmarking leads to inaccurate performance estimates. Here, we provide additional details on how we generated these synthetic inputs and examples of a few observed responses.

**Generation Logic**   We utilized the default random prompt generation logic from vLLM's benchmarking scripts, which are widely used in public benchmarks such as InferenceMax [4]. The following Python snippet illustrates the implementation:

```python
offsets = np.random.randint(
    0, tokenizer.vocab_size, size=num_prompts
)

for i in range(num_prompts):
    prompt = tokenizer.decode(
        [
            (offsets[i] + i + j) % tokenizer.vocab_size
            for j in range(int(self.input_len * 1.5))
        ]
    )

    re_encoded_sequence = tokenizer.encode(
        prompt, add_special_tokens=False
    )[:self.input_len]

    prompt = tokenizer.decode(re_encoded_sequence)
```

## E.1. Trivial Responses

This is the most common behavior. The model identifies the input as noise and produces a generic, sometime repetitive, comment. Because the draft model can easily predict this pattern, ARs are artificially high.

**Example Output** (Base: GPT-OSS 120b, Drafter: EAGLE3, Draft Length:3, Average AL: 3.44):

> "It looks like you've pasted a very long block of mixed-language text that doesn't form a clear question or request. I'm happy to help, but I need a bit more guidance. Could you let me know what you'd like to do with this text? For example: ..."

## E.2. Topic Latching

Occasionally, the random sampling produces a token or sub-sequence that the model interprets as a strong topical signal (e.g., a technical term or a common noun). The model "latches" onto this signal and generates a coherent, albeit seemingly random, response.

**Example Output** (Base: GPT-OSS 120b, Drafter: EAGLE3, Draft Length:3, Average AL: 1.877):

> "Below is an **expanded, production-ready roadmap** that takes you from the very first Unity install all the way to a **complete, polished 2-D platformer** (player, camera, enemies, collectibles, UI, audio, level loading, and a final build). Everything is broken into bite-size tasks, each with the exact actions you need to perform and ready-to-copy C# snippets. ..."

---

[4]https://github.com/InferenceMAX/InferenceMAX

## F. Tree-Based Verification Experiments

SPEED-Bench is designed not as a standalone inference framework, but rather as a lightweight evaluation wrapper that externalizes data processing, tokenization, and metric collection while integrating directly with production-grade serving engines such as SGLang, vLLM, and TensorRT-LLM. This design makes the framework inherently method-agnostic, enabling evaluation of any SD algorithm supported by the underlying backend implementation.

While the primary experiments in this work focus on draft-chain SD methods, which currently represent the dominant production deployment paradigm, the framework also fully supports tree-based speculative verification strategies. To demonstrate this flexibility, we conduct experiments using EAGLE3 tree-based verification with a Qwen3 235B target model in SGLang.

In these experiments, we vary both the branching factor (top-$k$) and the DL. Table 9 reports the resulting average AL measurements. As expected, increasing either the branching factor or the draft length improves AL by expanding the candidate verification space available to the target model.

*Table 9.* Average AL for tree-based SD using EAGLE3 with a Qwen3 235B target model in SGLang.

| top-$k$ \ **DL** | **3** | **5** | **7** |
|---|---|---|---|
| 2 | 2.55 | 2.81 | 2.89 |
| 4 | 2.82 | 3.19 | 3.31 |
| 8 | 3.04 | 3.49 | 3.65 |

# G. Expert Imbalance in Synthetic Benchmarking

MoE architectures rely on a gating network (router) to select a sparse subset of experts for each token. Since this router is trained on natural texts, when presented with random token inputs, which are statistically out-of-distribution, the router exhibits problematic behavior. It may "collapse" to a specific subset of experts, violating load-balancing assumptions of the inference engine. Consequently, the generation step time for MoE models differs between random noise and SPEED-Bench workloads, even in a standard autoregressive decoding setting.

Figure 11 illustrates the activation frequency of the top-k experts for a middle layer (Layer 17) in GPT-OSS 120B during the prefill of 8k ISL inputs at a batch size of 32. While SPEED-Bench inputs result in a relatively uniform activation profile, random tokens lead to significant imbalance, where the router disproportionately favors a subset of experts.

Figure 12 tracks the total number of unique experts activated across layers of the model. Notably, processing random tokens fails to activate **20-30%** of available experts in certain layers. This lack of coverage is interesting given the high volume of tokens ($32 \times 8000$), confirming that synthetic noise fails to trigger the routing logic that occurs on real semantic workloads.

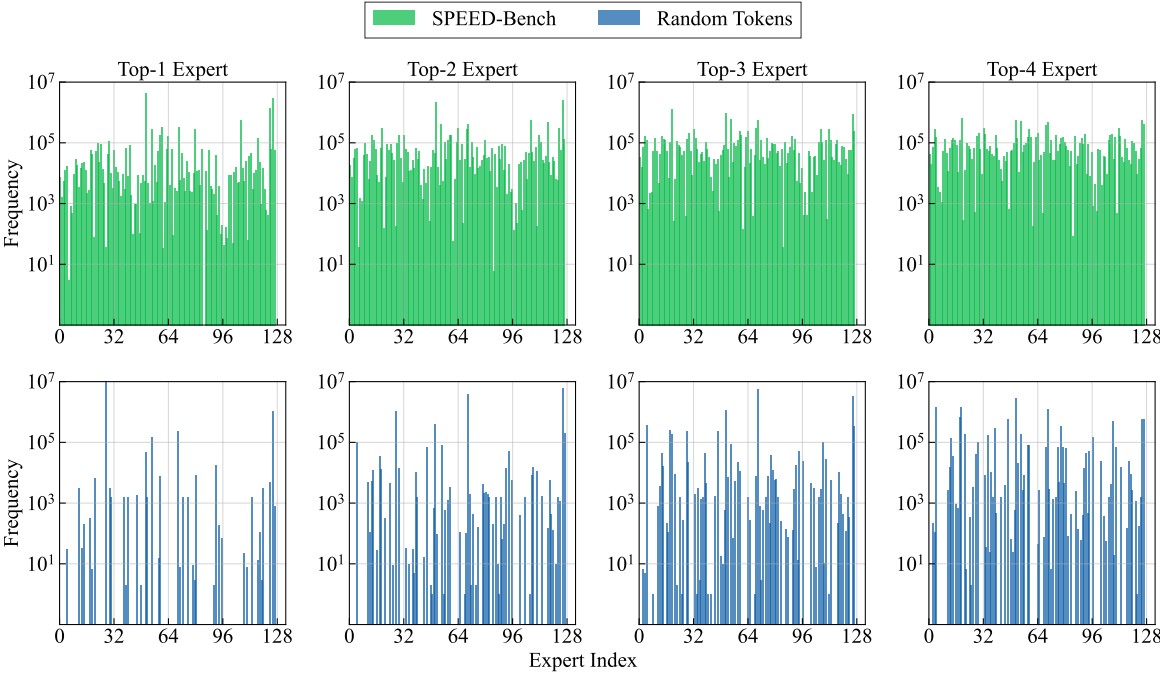

*Figure 11.* Distribution of the top four activated experts of GPT-OSS 120B (17th layer) during the prefill stage. Horizontal axis indicates the expert index (1 to 128). Top plots are using the Throughput Split (8k), bottom plots use random input tokens. BS=32.

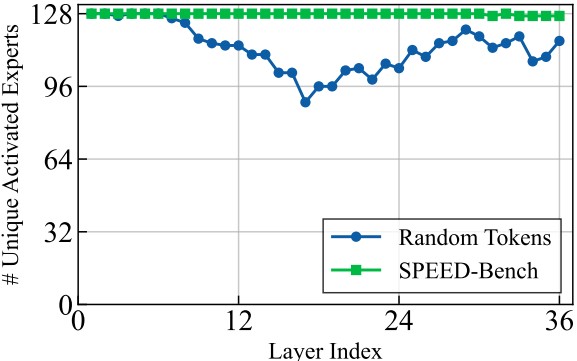

*Figure 12.* Number of unique experts activated as a function of layer index, comparing random input tokens to the Throughput Split (8k). Target model is GPT-OSS 120B, BS=32.

## H. Estimating Domain-Specific Speedups using the Throughput Split

The Throughput Split is designed to provide measurements of system efficiency under realistic loads. A practical benefit of this design is that it enables practitioners to estimate the expected speedup using SD for fine-grained domains without constructing and performing exhaustive throughput benchmarks for every specific category.

We define the speedup $S$ as the ratio of the effective token generation speed of SD to that of the baseline autoregressive system. Let $t_{ar}$ be the average time required for a single decoding step in the baseline autoregressive system (generating exactly 1 token per step). Let $t_{sd}$ be the average time required for a single decoding step in SD (generating on average $AL$ tokens per step).

The speedup is derived as follows:

$$S = \frac{\text{Tokens}_{sd}/\text{Time}_{sd}}{\text{Tokens}_{ar}/\text{Time}_{ar}} = \frac{AL/t_{sd}}{1/t_{ar}} = \frac{t_{ar} \cdot AL}{t_{sd}} \tag{3}$$

Intuitively, the decoding step latencies ($t_{ar}$, $t_{sd}$) are primarily determined by the serving system configuration (e.g., engine implementation, ISL bucket, BS, and hardware platform), whereas the average accepted length (AL) is predominantly domain-dependent. This separation allows practitioners to combine throughput measurements collected on one representative workload with AL measurements collected on another target domain. This allows for the estimation of domain-specific speedups by combining the system-dependent latency metrics ($t_{ar}$, $t_{sd}$) with the domain-dependent AL.

### H.1. Protocol

To estimate the speedup for a specific domain (e.g., Gaming) **at a specific serving scenario** (e.g., $BS = 32$, ISL 8k):

1. **Measure Step Times ($t_{ar}, t_{sd}$):** Use our measurement framework and the Throughput Split (specifically the bucket matching your target ISL) at the desired BS.

   - $t_{ar}$: The average "time per step" measured for standard autoregressive decoding.
   - $t_{sd}$: The average "time per step" measured for SD at a selected draft length (DL).

2. **Measure average AL:** Using a set of indicative prompts representing the target domain, measure the AL with the selected DL.

3. **Calculate:** Plug the measured values into Equation 3 to obtain the approximated speedup.

### H.2. The Necessity of Realistic Data

Crucially, this proxy method relies on accurate measurements of $t_{ar}$ and $t_{sd}$. Using random token inputs (a common practice for throughput testing) yields inaccurate step time measurements for SD (Section 8.4), and even for baseline autoregrssive decoding on MoE models due to expert imbalance (Appendix G). Because the Throughput Split utilizes genuine semantic data, it avoids these artifacts, ensuring that $t_{ar}$ and $t_{sd}$ are reliable.

### H.3. Validation of Proxy Measurements for Estimating Speedups

To validate the proxy measurements for estimating speedups, we compare projected speedups computed using Equation 3 against directly measured end-to-end speedups across three ISL buckets (1k, 2k, and 8k) of the Throughput Split. We first measure the step latencies ($t_{ar}$, $t_{sd}$) using the *Mixed* category (STEM and QA problems), and subsequently use the measured AL values from two distinct domains: a *High Entropy* category (Creative Writing) and a *Low Entropy* category (Coding and Sorting problems). Experiments are conducted using a Llama 3.3 70B target model with $BS = 16$ and $DL = 3$.

As shown in Table 10, the projected speedups closely match the measured end-to-end speedups across both EAGLE3 and Vanilla SD methods.

*Table 10.* Validation of proxy-based speedup estimation using the Throughput Split. Step latencies $(t_{ar}, t_{sd})$ are measured on the Mixed category and combined with domain-specific AL measurements to project SD speedups. Results closely match the directly measured end-to-end speedups across both entropy regimes and ISLs. Experiments use Llama 3.3 70B target model with $BS = 16$ and $DL = 3$.

| Drafter | Target Category | ISL | AL | $t_{ar}$ (ms) | $t_{sd}$ (ms) | Actual | Projected |
|---------|-----------------|-----|------|------|------|--------|-----------|
| EAGLE3  | High Entropy | 1k | 2.12 | 19.6 | 25.9 | 1.63× | 1.61× |
|         | High Entropy | 2k | 1.86 | 20.4 | 27.5 | 1.39× | 1.38× |
|         | High Entropy | 8k | 1.17 | 25.2 | 34.1 | 0.85× | 0.86× |
|         | Low Entropy  | 1k | 2.93 | 19.6 | 25.9 | 2.23× | 2.22× |
|         | Low Entropy  | 2k | 2.59 | 20.4 | 27.5 | 1.91× | 1.92× |
|         | Low Entropy  | 8k | 1.19 | 25.2 | 34.1 | 0.87× | 0.88× |
| Vanilla | High Entropy | 1k | 2.47 | 19.7 | 35.3 | 1.37× | 1.38× |
|         | High Entropy | 2k | 2.49 | 20.4 | 38.6 | 1.31× | 1.32× |
|         | High Entropy | 8k | 2.60 | 25.2 | 59.9 | 1.09× | 1.09× |
|         | Low Entropy  | 1k | 3.12 | 19.7 | 35.3 | 1.74× | 1.74× |
|         | Low Entropy  | 2k | 3.16 | 20.4 | 38.6 | 1.67× | 1.67× |
|         | Low Entropy  | 8k | 3.21 | 25.2 | 59.9 | 1.36× | 1.35× |

# I. Detailed Experimental Setup

We describe in detail the experimental setup, including checkpoints, engines, and settings, in our experiments (Section 8). Unless explicitly stated otherwise, all results use greedy decoding (Temperature=0).

**Checkpoints** We utilize publicly available target ( Table 11) and draft models (Table 12).

*Table 11.* Target model checkpoints used in the experiments, including Tensor Parallel (TP) and Expert Parallel (EP) configurations. For GPT-OSS 120B we use the default reasoning effort (*medium*) except for Section 8.2 where we used *low*.

| Target Model | HuggingFace Repo / Source | TP | EP |
|---|---|---|---|
| GPT-OSS 120B (OpenAI, 2025) | openai/gpt-oss-120b | 1 | 1 |
| Llama 3.3 70B (Grattafiori et al., 2024) | meta-llama/Llama-3.3-70B-Instruct | 1 | 1 |
| Qwen3-Next (Yang et al., 2025a) | Qwen/Qwen3-Next-80B-A3B-Instruct | 8 | 8 |
| Qwen3 235B (Yang et al., 2025a) | Qwen/Qwen3-235B-A22B | 8 | 8 |
| DeepSeek R1 (Guo et al., 2025) | deepseek-ai/DeepSeek-R1 | 8 | 4 |

*Table 12.* Draft model checkpoints used in SPEED-Bench evaluation. (*) Used exclusively in Appendix N.

| Target Model | SD Algorithm | HuggingFace Repo / Source |
|---|---|---|
| GPT-OSS 120B | EAGLE3 | nvidia/gpt-oss-120b-Eagle3-long-context |
| GPT-OSS 120B (*) | EAGLE3 | lmsys/EAGLE3-gpt-oss-120b-bf16 |
| Llama 3.3 70B | EAGLE3 | yuhuili/EAGLE-LLaMA3-Instruct-70B |
| Llama 3.3 70B | Vanilla | meta-llama/Llama-3.2-1B-Instruct |
| Qwen3-Next | EAGLE3 | lmsys/SGLang-EAGLE3-Qwen3-Next... |
| Qwen3 235B | EAGLE3 | nvidia/Qwen3-235B-A22B-Eagle3 |
| Qwen3 235B | Vanilla | Qwen/Qwen3-0.6B |

**Inference Engines** We use three inference engines in our experiments: TensorRT-LLM, vLLM and SGLang. For each engine, we use the official Docker images provided by the respective framework, with version details reported in Table 13. TensorRT-LLM 1.2.0rc7 is used exclusively for results in Table 1 requiring temperature sampling ($T=1$) with non-vanilla SD, as this setup is not supported in rc1.

*Table 13.* Engine versions and Docker images. (*) Used exclusively in Table 1 with $T=1$ for non-vanilla SD.

| Engine | Docker Image |
|---|---|
| TensorRT-LLM | nvcr.io/nvidia/tensorrt-llm/release:1.2.0rc1 |
| | nvcr.io/nvidia/tensorrt-llm/release:1.2.0rc7 (*) |
| SGLang | lmsysorg/sglang:v0.5.7 |
| vLLM | vllm/vllm-openai:v0.13.0 |

**EAGLE3 Training Configuration** We train several EAGLE3 models for GPT-OSS 120B, specifically for Section 8.2 and Section 8.5. We use NVIDIA's Model-Optimizer training framework [5], and select the Nemotron Post-Training Dataset V2 [6] as a training data source. We sample 500k training samples with uniform weight for all samples except for those from multilingual categories which have 10x lower weight due to their over-representation in the dataset. We employ synthetic generation to regenerate the responses using the target model with randomized reasoning effort and temperature (between 0 and 1) for each generation. All models are trained for 3 epochs using the AdamW optimizer with a cosine-scheduled learning rate of $3 \cdot 10^{-4}$ and a minimum learning rate of $1 \cdot 10^{-4}$. We train on 8xB200 GPUs with an effective training batch size of 128. Training sequence length varies per experiment, with the models in the vocabulary pruning experiment using sequence length 1024. RoPE scaling is configured as described in Appendix N.

---

[5]https://github.com/NVIDIA/Model-Optimizer

[6]https://huggingface.co/datasets/nvidia/Nemotron-Post-Training-Dataset-v2

## J. Measuring ALs on the Throughput Split

We analyze the stability of speculation accuracy across varying ISLs using the fixed buckets of the Throughput Split. The goal of this experiment is to act as a sanity check for our data categorization, verifying that *Low Entropy* samples indeed yield higher ALs than *High Entropy* samples.

Figure 13 presents the average AL as a function of ISL for three setups. For Vanilla SD (Llama 3.3 70B) and Native MTP (Qwen3-Next), we observe the expected behavior: *Low Entropy* prompts (e.g., coding, sorting) yield the highest ALs. *High Entropy* prompts (e.g., creative writing, roleplay) yield the lowest ALs. *Mixed Entropy* prompts (e.g., STEM and general knowledge) fall in between. Furthermore, these methods demonstrate stability, with ALs remaining relatively constant as the ISL grows.

In contrast to the behavior mentioned above, the GPT-OSS 120B with EAGLE3 setup exhibits an anomaly. While it follows the expected trend at short contexts (1k), the AL for the *Low Entropy* category degrades as the ISL increases, crossing below the *Mixed Entropy* curve. We attribute this degradation to the training distribution of the specific EAGLE3 draft model, which heavily favors general knowledge prompts over structured coding tasks. The model was trained on a combination of UltraChat (Ding et al., 2023) and Magpie-Llama-3.1-Pro-300K (Xu et al., 2024) [7]. A closer look reveals a scarcity of code in these sources. UltraChat is divided into three sectors: the first contains 30 topics focused on generic concepts and questions, with no coding content. The remaining two sectors contain 20 topics each, yet only a single topic in each sector is dedicated to coding. Similarly, the Magpie dataset contains less than 8% coding samples overall. Additionally, this behavior may be exaggerated by incorrect RoPE scaling configuration, as discussed in Appendix N.

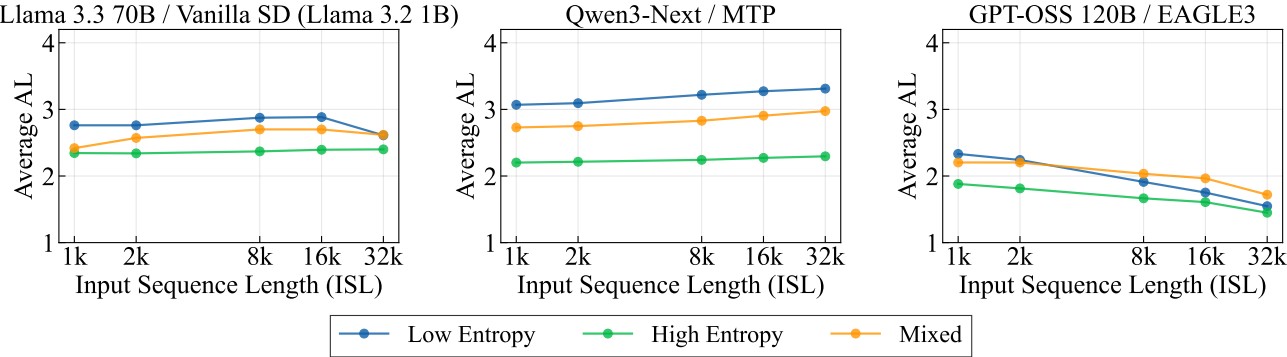

*Figure 13.* AL Stability across ISLs. AL measured on the Throughput Split buckets (1k–32k) for three different setups.

## K. Vocabulary Pruning Analysis

We further analyze vocabulary pruning by conducting a theoretical token analysis using completions with greedy sampling generated from the Qualitative Split, and our EAGLE3 training corpus as the reference dataset (see Appendix I). By filtering to the top $K$ most frequent tokens from the training corpus, as done in EAGLE3, we establish an upper bound on the achievable AR on the test set. As demonstrated in Table 14, aggressive vocabulary pruning results in a marginal reduction in token coverage for the overall test set. However, we observe that specific sub-domains, particularly Multilingual data, are disproportionately sensitive to this reduction.

*Table 14.* Percentage of SPEED-Bench output tokens present in reduced vocabulary.

| K | GPT-OSS Reasoning: *Low* | | GPT-OSS Reasoning: *Medium* | |
|---|---|---|---|---|
| | Overall | Multilingual | Overall | Multilingual |
| 16k | 89.7% | 72.1% | 89.1% | 73.9% |
| 32k | 94.7% | 76.9% | 94.5% | 78.1% |
| 64k | 98.3% | 82.2% | 98.2% | 83.1% |
| Full | 100.0% | 98.9% | 100.0% | 98.9% |

---

[7]https://huggingface.co/datasets/Magpie-Align/Magpie-Llama-3.1-Pro-300K-Filtered

## L. Extended SpecBench vs SPEED Results

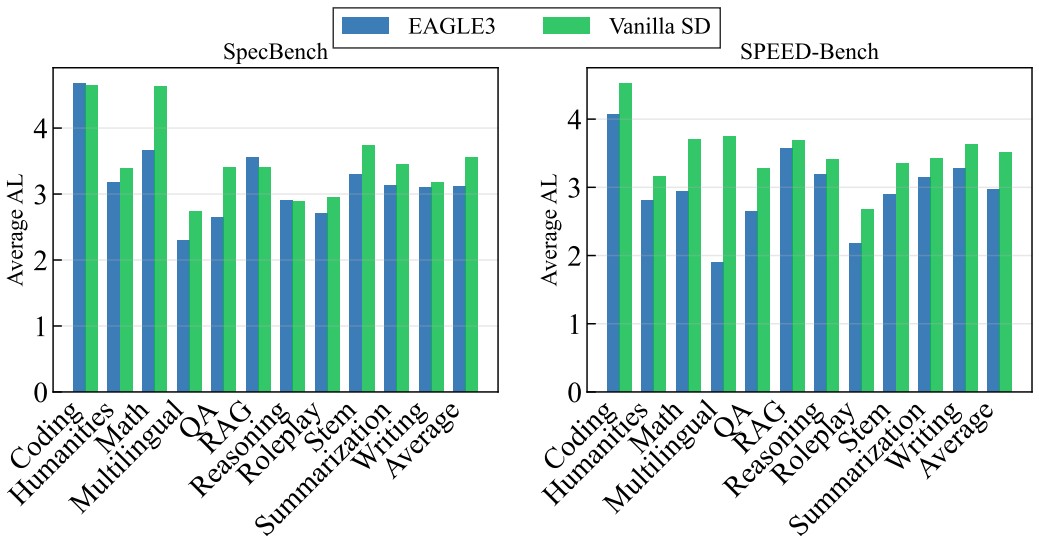

*Figure 14.* Average AL across all categories in SpecBench vs. SPEED-Bench. Target model is Llama 3.3 70B. DL=7, BS=32.

## M. Inference Engine Comparison

In Section 8.4, we briefly discussed the performance differences between inference backends. Here we provide the full comparison between TensorRT-LLM and vLLM.

Figure 15 compares the throughput of TensorRT-LLM and vLLM. Both frameworks are orchestrated in Python, which can introduce host synchronization overhead and kernel launch latency compared to C++ implementations. To mitigate this, both engines leverage CUDA Graphs to capture and replay device operations with a single launch. We observe that TensorRT-LLM achieves higher throughput in this configuration, largely due to its support for a *one-model* runtime paradigm. In this setup, the speculative head is appended directly to the target model, enabling a single CUDA Graph to capture the entire verification and drafting loop. In contrast, vLLM utilizes a *two-model* approach (also supported by TensorRT-LLM) where the draft model runs as a separate engine. This separation naturally incurs additional host overhead due to inter-engine communication, although mechanisms like async/overlap scheduling help hide this latency. We note that vLLM's piecewise graph construction may offer greater flexibility for dynamic drafting strategies by reducing static shape requirements.

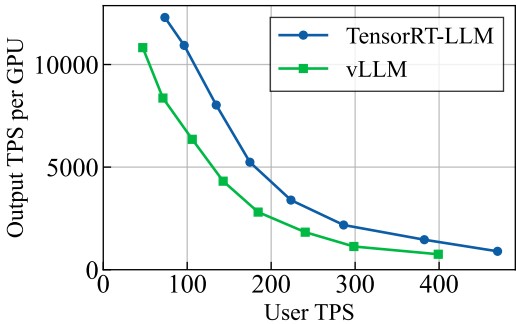

*Figure 15.* Throughput as a function of user TPS, comparing different engines. Target is GPT-OSS 120B with EAGLE3 drafter, measured on the Throughput Split ($2k$). DL=2. Points represent BS from 2 to 256.

# N. Long-Context Inaccuracy in Existing EAGLE3 Models

During experimentation with the Throughput Split, we identified unexpected degradation in two EAGLE3 models on HuggingFace when evaluating at higher ISL buckets. Based on our results, one possible explanation could be incorrect configuration of RoPE scaling. We examine the following EAGLE3 heads for GPT-OSS 120B: lmsys/EAGLE3-gpt-oss-120b-bf16 and nvidia/gpt-oss-120b-Eagle3-long-context. Figure 16 compares both of these models as well as a reference checkpoint we trained using 4k context length and setting YaRN scaling for inference.

Examining the former checkpoint, we notice that RoPE scaling is not configured which indicates that the model will not perform well during inference when the ISL exceeds the maximum ISL used during training. Based on the results in the figure, we hypothesize it was trained with a maximum context length of 8k, sufficient for many tasks but suffering in inference as the ISL grows beyond its supported range.

For the latter checkpoint, we observe decay even for 8k ISL, which is surprising considering the RoPE scaling configuration value of *original_max_position_embeddings* is set to 8192. While the exact cause of this discrepancy is unclear, one explanation could be that the default value for Llama3 (which is 8192) was not changed during training, and does not reflect the actual context length used during training.

Given these results, we recommend training EAGLE3 with *max_position_embeddings* equal to the training context length, and applying RoPE scaling techniques in the inference configuration. For optimal results, a long-context fine-tuning phase could be introduced as recommended by (Peng et al., 2024).

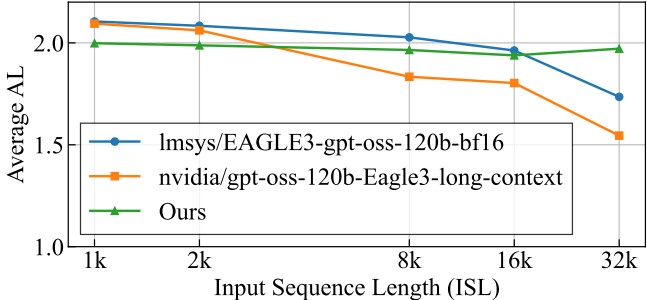

*Figure 16.* AL Stability across various models. Average AL measured on the Throughput Split buckets (1k–32k). Target is GPT-OSS 120B, with three EAGLE3 drafters. Carefully configured RoPE scaling can ensure stability over all context lengths.

