# OpenReview forum: "SPEED-Bench: A Unified and Diverse Benchmark for Speculative Decoding"
_ICML.cc/2026/Conference — ICML 2026 regular_

### Official Review · Reviewer_tjDV · 2026-03-09

**Soundness:** 3
**Presentation:** 3
**Significance:** 3
**Originality:** 3
**Overall Recommendation:** 4
**Confidence:** 4

**Summary:**

This paper proposes SPEED-Bench, a benchmark for evaluating speculative decoding. The motivation is that existing works lack both task diversity and throughput-oriented evaluation. SPEED-Bench includes two evaluation splits: a qualitative split, which covers 11 diverse task types, and a throughput split, which examines performance under varying serving parameters, such as batch size and sequence length. The benchmark also incorporates a semantic-orthogonality selector to choose representative data items.

**Compliance With Llm Reviewing Policy:**

Affirmed.

**Final Justification:**

The rebuttal has successfully resolved my concerns. I am leaning towards acceptance and will maintain my positive score of 4, as the contribution is solid.

**Key Questions For Authors:**

1.	What is the method of difficulty measurement for the questions in the qualitative split?

2.	Which algorithm is used for sequence-length alignment in batched SD?

3.	How about conducting an ablation study about Local Swap Refinement, to further demonstrate its effectiveness?

**Limitations:**

I have no specific concerns about the limitations.

**Strengths And Weaknesses:**

Strengths:

The proposed dataset emphasizes task diversity, addressing a core limitation of existing benchmarks such as SpecBench.

The semantic orthogonality ensured by the Local Swap Refinement in Algorithm 1 is crucial, further enhancing item-level diversity beyond the task level.

Evaluation is performed across various model combinations (e.g., SD, EAGLE-3), strengthening the validity of the results.

Integration with current inference engines (vLLM and TensorRT, as claimed) improves reproducibility.

Weaknesses:

1.	As claimed in line 193, the qualitative split provides a difficulty field for Coding, Humanities, Math, and STEM. Yet, how to measure the difficulty of the questions is not specified.

2.	The algorithm of SD in batch size=1 is clear, while it is not when batch size > 1. There should be an approach to handle the accept sequences to ensure matrix-dimension alignment across batches (the length of different batches is different from each other), while the detail has not been specified.

3.	TTFT is used as a metric for performance, as stated in sec7, but SD is not essentially used in the prefilling stage, but in the decoding stage. Therefore, the usage of this metric is confusing.

4.	Despite the success of Local Swap Refinement, the ablation about it is missing.

Minor concerns:

5.	The use of abbreviations may be excessive (e.g., AL for accept length, ISL for input sequence length). I recommend using simple symbols (e.g., $n$ for length) to improve readability.

---

> ### Author Rebuttal · Authors · 2026-03-30
>
> We thank the reviewer for the positive feedback, the constructive notes, and for recognizing the value of SPEED-Bench's task diversity and engine integration. We address your specific questions and weaknesses below:
>
> * **Weakness 1 + Question 1** (difficulty labels are not specified): We assign difficulty levels for Coding, Humanities, Math, and STEM according to the following schema:
>     * Math, STEM, and Humanities: MT-Bench samples are labeled "medium", while Humanity’s Last Exam samples are labeled "hard". Additionally, for Math, GSM8K samples are labeled as "easy".
>     * Coding: MT-Bench samples are labeled "medium". For LiveCodeBench, we use the difficulty labels provided by the benchmark. For CodeContests, we adopt the benchmark difficulty and normalize it by mapping "hard", "harder", and "hardest" to "hard", and converting levels "A-I" to "easy", "J-Q" to "medium", and "R-Z" to "hard".
> We apologize if this was not sufficiently clear in the text, and we will explicitly clarify how these labels are sourced from the upstream datasets in the final version.
>
> * **Weakness 2 + Question 2** (batched speculative decoding explanation): The reviewer is correct that different sequences within a batch will accept a different number of tokens. However, in modern inference engines, this divergence does not actually require complex matrix-dimension alignment or padding during the forward passes. Instead, the input shape to the target model remains constant across the batch: $Batch \times (num\textunderscore draft\textunderscore tokens + 1)$. The "+1" accounts for the previous verified "free" token, which still needs its KV cache generated by the target model. **The divergence in accepted lengths is handled entirely through KV cache management**. If a sequence rejects a draft token, the system simply "rolls back" the KV cache of the drafter to the exact point of the last accepted token. The drafter then prefills the new verified "free" token and begins the next drafting phase. Because of this rollback mechanism, the input shapes for the next forward pass remain uniformly aligned across the batch, regardless of how many tokens each individual sequence accepted in the previous step. We agree this is valuable context for readers, and we will add an explanation of this mechanism to the appendix in the final revision.
>
> * **Weakness 3** (confusing usage of TTFT metric): The reviewer is correct that Speculative Decoding is inherently a decode-phase optimization and does not accelerate the prefilling stage. We log Time To First Token (TTFT) in our measurement framework strictly to provide a holistic, end-to-end picture of the serving metrics for practitioners who need to monitor full system performance. We will add a clarifying note in the text to ensure readers do not mistakenly infer that SD improves TTFT.
>
> * **Weakness 4 + Question 3** (ablation on Local Swap Refinement step in selection algorithm): We appreciate the suggestion to ablate the Local Swap Refinement step. We measured the average pairwise cosine similarity of our subsets using the greedy algorithm both with and without the swap refinement (lower similarity indicates higher diversity). As shown below, the swap refinement is highly effective, improving diversity by 150% in the Multilingual category and 125% in Coding. We will include this ablation study in the final paper revision.
> |Category|SPEED-Bench|Greedy w/o Local Swap Refinement|
> |---|---|---|
> |Math|0.15|0.17|
> |Coding|0.16|0.36|
> |Multilingual|0.06|0.15|
> |Writing|0.18|0.19|
> |Humanities|0.12|0.14|
> |STEM|0.12|0.15|
> |Reasoning|0.14|0.14|
> |RAG|0.13|0.15|
> |Roleplay|0.24|0.24|
> |Summarization|0.11|0.11|
> |QA|0.10|0.10|
>
> * **Weakness 5** (abbreviations): We agree that the text is dense with abbreviations. We will attempt to streamline the terminology in the final revision, replacing acronyms with clearer symbols or fully spelled-out terms where appropriate.

---

> > ### Author Rebuttal · Reviewer_tjDV · 2026-04-01
> >
> > Thank you for preparing the response, which has thoroughly addressed my concerns. I think my current positive score is a fair evaluation, and I will maintain my score of 4.

---

> > > ### Author Response · Authors · 2026-04-05
> > >
> > > Dear reviewer,
> > >
> > > Thank you again for your time, your constructive feedback, and for confirming that our response thoroughly addressed your concerns. We truly appreciate your engagement and your positive recommendation for the paper's acceptance.
> > >
> > > We respect your perspective that a 4 represents a fair evaluation of the paper as it stands. Since our goal is to make this work as strong as possible, we would love to know if there are any open issues or weaknesses we missed, as we would be happy to put in the work to address them.
> > >
> > > Best,
> > > The Authors

---

### Official Review · Reviewer_BAEv · 2026-03-12

**Soundness:** 2
**Presentation:** 2
**Significance:** 2
**Originality:** 2
**Overall Recommendation:** 4
**Confidence:** 3

**Summary:**

This paper introduces SPEED-Bench, a benchmark suite for speculative decoding that combines two data splits with different purposes: a Qualitative Split for measuring acceptance-related metrics across semantically diverse prompt categories, and a Throughput Split for measuring latency and throughput under realistic concurrency and input-length regimes. The benchmark is paired with a measurement framework that interfaces with production-grade engines such as vLLM, TensorRT-LLM, and SGLang while externalizing tokenization and prompt formatting for consistency. The paper also presents empirical case studies showing how benchmark choice affects measured acceptance length and throughput, including comparisons against SpecBench, analyses of synthetic versus real prompts, vocabulary pruning effects, and long-context behavior.

**Compliance With Llm Reviewing Policy:**

Affirmed.

**Final Justification:**

I have no concerns.

**Key Questions For Authors:**

1. Can you provide a direct validation of the Section 6 proxy for estimating domain-specific speedups? In particular, how well do estimated speedups match measured speedups across several held-out domains, batch sizes, and ISLs? Strong evidence here would increase my confidence noticeably.

2. How much of the behavior in Figure 7 remains after removing the (BS>256) region or otherwise controlling for the Python asyncio / GIL overhead acknowledged on Page 5? If the draft-length crossover still appears cleanly in a measurement regime you consider trustworthy, that would strengthen a key claim.

3. Did you test whether random subsets from the same candidate pool lead to less stable or less discriminative conclusions than the curated Qualitative Split? This is important because Figure 2 shows lower similarity, but the paper does not yet show that the selection algorithm itself improves benchmarking outcomes.

**Limitations:**

1. The selection algorithm is not validated against a strong enough alternative:
The paper compares the curated set against SpecBench and against random selection in terms of similarity, but not in terms of downstream benchmarking quality. A key missing experiment is whether several random subsets from the same source pool produce noisier rankings, weaker discrimination, or less stable conclusions than the curated subset. Without that, the benchmark may be good because the source pool is good, not because the algorithm is particularly meaningful.

2. The main benchmark claim is broader than the demonstrated scope:
The paper presents SPEED-Bench as a unified SD benchmark, but Page 6 states that experiments focus exclusively on draft chains rather than tree-based verification. That is a narrower demonstrated scope than the framing suggests. If the current engines or serving constraints motivate that choice, the framing should be correspondingly more precise.

3. A central practical claim, the speedup proxy from Section 6, is not validated: The paper proposes that one can estimate domain-specific speedups by combining measured step times with measured AL. Algebraically this is trivial, but scientifically the important assumption is that step latency is largely system-dependent and sufficiently transferable across domains. The paper does not test this. This matters because it is one of the main practical selling points of the Throughput Split.

**Strengths And Weaknesses:**

1. The paper tackles a real and important evaluation gap. Speculative decoding is very sensitive to workload characteristics, so benchmark quality matters a lot more here than in many standard inference papers.

2. The split between a semantic-quality benchmark and a serving-oriented throughput benchmark is sensible and well motivated. This is one of the stronger design choices in the paper.

3. Integration with production engines is practically valuable. For this topic, benchmarking only in high-level research code is often too optimistic, and the paper is right to push evaluation closer to deployment reality.

---

> ### Author Rebuttal · Authors · 2026-03-29
>
> We sincerely thank the reviewer for their rigorous review and the push to validate the claims on the speedup proxy and the selection algorithm. We provide additional experiments below to strengthen these claims and the paper:
>
> **Question 1 & Limitation 3** (validation of the proxy for estimating speedups) - The fact that step latency is largely system-dependent and thus transferable, allows to use the Throughput Split for speedup estimations. To validate this, we measured the actual vs. projected speedups across three ISL buckets (1k, 2k, 8k). We used the Mixed category (STEM/QA) to measure the step times ($t_{ar}$, $t_{sd}$). Then, we project the speedups for the High Entropy (Creative Writing) and Low Entropy (Coding/Sorting) categories using their respective ALs. As shown below (Llama 3.3 70B target, BS=16, DL=3), the projections match the measured speedups very well. Other batch sizes (8,32) show similar behavior but omitted due to rebuttal char limit.
>
> |Drafter|Target Cat.|ISL|AL|t_ar(ms)|t_sd(ms)|Actual|Proj.|
> |---|---|---|---|---|---|---|---|
> |**EAGLE3**|High Entropy|1k|2.12|19.6|25.9|1.63x|1.61x|
> |||2k|1.86|20.4|27.5|1.39x|1.38x|
> |||8k|1.17|25.2|34.1|0.85x|0.86x|
> ||Low Entropy|1k|2.93|19.6|25.9|2.23x|2.22x|
> |||2k|2.59|20.4|27.5|1.91x|1.92x|
> |||8k|1.19|25.2|34.1|0.87x|0.88x|
> |**Vanilla**|High Entropy|1k|2.47|19.7|35.3|1.37x|1.38x|
> |||2k|2.49|20.4|38.6|1.31x|1.32x|
> |||8k|2.60|25.2|59.9|1.09x|1.09x|
> ||Low Entropy|1k|3.12|19.7|35.3|1.74x|1.74x|
> |||2k|3.16|20.4|38.6|1.67x|1.67x|
> |||8k|3.21|25.2|59.9|1.36x|1.35x|
>
> **Question 2** (robustness of Figure 7 after removing BS>256) - As mentioned in the paper, Python overhead at extreme batch sizes (>256) is possible. However, the optimal draft length crossover **occurs before** this region. Figure 7 shows that DL=1,3 are comparable at BS=128. By BS=256, DL=1 outperforms DL=3, and the baseline (no specdec) is closer to DL=3. We acknowledge that the BS=512 point is confusing, and we will remove it from the final revision. We are also actively working on an *optimal* async solution. To further validate our claim, we replicate this experiment using `vllm-bench` and observed similar crossover trends:
>
> |BS|DL=1 (uTPS/oTPS)|DL=3 (uTPS/oTPS)|Base (uTPS/oTPS)|
> |---|---|---|---|
> |64|97.9/6.1k|**100.7/6.2k**|85.0/5.3k|
> |128|**77.2/9.3k**|71.9/8.5k|64.2/7.7k|
> |256|**55.0/12.5k**|46.7/10.6k|50.6/11.6k|
> |512|**35.8/14.9k**|29.1/12.1k|**36.1/15.1k**|
>
>
> **Question 3 & Limitation 1** (impact of the selection algorithm vs. random subsets) - We thank the reviewer for this suggestion. We conducted the experiment suggested and will update the paper with these results. We generated 10 independent uniform random subsets and 10 subsets using our selection algorithm (different seeds), all from the same source pool, maintaining the same number of samples per category. We evaluated each subset using Llama 70B and Eagle3.
>
> * *Unstable per-category measurements under uniform sampling*
> Under uniform sampling, per-category AL measurements vary substantially across subsets. The table below compares the relative range (Range% = (max−min)/mean) between uniform and optimized subsets at DL=7:
> |Cat.|Uniform|Ours|
> |---|---|---|
> |roleplay|22.7%|**0.6%**|
> |multilingual|14.6%|**5.1%**|
> |rag|12.1%|**0.9%**|
> |humanities|9.2%|**3.0%**|
> |writing|7.2%|**0.9%**|
> |stem|6.5%|**2.8%**|
> |math|5.6%|**1.6%**|
> |summarization|4.4%|**2.0%**|
> |coding|4.0%|**0.7%**|
> |reasoning|3.9%|**2.7%**|
> |qa|2.2%|2.6%|
>
> * *Inconsistent category rankings* A reliable benchmark should rank categories consistently (conclusions on which domains are "easy"/"hard" for a speculator should not depend on the random sample drawn). We measure ranking agreement using the normalized Kendall-tau distance (fraction of pairwise orderings that disagree between two subsets).
> At DL=7, our selection algorithm reduces mean ranking disagreement by 14X compared to uniform sampling. Concretely, 8 out of 9 optimized subsets produce the identical category ranking, while only 2 out of 10 uniform subsets share the same ranking (8 distinct rankings among 10 uniform subsets).
> |Config|Mean K-τ|Max K-τ|
> |---|---|---|
> |Uniform DL=3|0.046|0.091|
> |Optimal DL=3|**0.019**|**0.055**|
> |Uniform DL=7|0.056|0.127|
> |Optimal DL=7|**0.004**|**0.018**|
>
>
> **Limitation 2** (tree-based speculation) - We designed the measurement framework as a wrapper that handles data and statistic gathering externally while integrating directly with production-grade engines. Because of this, our benchmark is method-agnostic: whatever algorithms the underlying backends support, we can run. While our primary experiments focused on draft chains (the production standard for BS > 1), the framework natively supports tree-based speculation. To demonstrate this, we ran evaluations for Qwen3 235B/Eagle3 in SGLang. We measure AL, vary the branching factor (top-k) from 2 to 8 and the draft length from 3 to 7:
> |top-k\DL|3|5|7|
> |---|---|---|---|
> |2|2.55|2.81|2.89|
> |4|2.82|3.19|3.31|
> |8|3.04|3.49|3.65|

---

> > ### Author Rebuttal · Reviewer_BAEv · 2026-04-01
> >
> > The authors resolved my concerns and I have raised my score. Good luck.

---

> > > ### Author Response · Authors · 2026-04-05
> > >
> > > Dear reviewer,
> > >
> > > Thank you again for your time, your constructive feedback and suggestions, and for raising the initial score.
> > > We truly appreciate your engagement and your positive recommendation for the paper's acceptance.
> > >
> > > Best,
> > > The authors

---

### Official Review · Reviewer_NJCC · 2026-03-13

**Soundness:** 3
**Presentation:** 3
**Significance:** 3
**Originality:** 3
**Overall Recommendation:** 4
**Confidence:** 3

**Summary:**

This paper proposes SPEED-Bench, an evaluation suite for speculative decoding approaches. Unlike existing evaluation frameworks such as Spec-Bench, SPEED-Bench aims to improve the diversity and coverage of evaluation tasks, while also covering different evaluation setups including sequential decoding and batch decoding. The benchmark consists of a Qualitative data split with maximized diversity and a Throughput data split for evaluating concurrent generation.

**Compliance With Llm Reviewing Policy:**

Affirmed.

**Final Justification:**

I recommend acceptance of this paper, due to the reasons outlined in the "Strengths And Weaknesses" section. While there was a minor weakness, the authors successfully added clarifications and resolved it during the rebuttal period.

**Key Questions For Authors:**

[Q1] How does pre-tokenization ensure consistent evaluation across multiple serving frameworks such as vLLM and SGLang?

**Limitations:**

While the authors did not mention any potential negative societal impact of the work, I do not see any potential social impact that needs to be discussed.

**Strengths And Weaknesses:**

**Strengths**

[S1] The paper is clearly written and easy to follow.

[S2] The proposed benchmark suite is very practical and covers areas of speculative decoding evaluation that were previously difficult to measure with existing benchmarks.

[S3] The paper clearly outlines the methodology used to construct a high-diversity dataset.

[S4] The paper includes thorough analysis, especially in the appendix section, that provides broader insights, more than simply gathering an evaluation dataset.

**Weaknesses**

[W1] It is unclear if pre-tokenization ensures consistent evaluation across multiple serving frameworks such as vLLM and SGLang. For example, many factors beyond tokenizer mismatch (e.g., numerical differences arising from floating point operations such as matrix multiplication ordering) could affect the final outcome. Therefore, pre-tokenization alone may not be sufficient to ensure consistent evaluation across different serving frameworks.

---

> ### Author Rebuttal · Authors · 2026-03-29
>
> We thank the reviewer for the positive feedback, the constructive notes, and for recognizing the practical value and thoroughness of SPEED-Bench.
>
> The reviewer raises a great point. It is true that evaluating the exact same prompt on different inference engines will often yield different generated responses due to numerical variations (e.g., floating-point non-determinism, matrix multiplication ordering, or specific kernel implementations). To address this, we draw a strict distinction between external preprocessing factors and internal engine factors:
> * **External factors (tokenization & prompting):** Different engines often have different ways of applying chat templates or injecting/omitting special tokens. If Engine A prepends a different special token than Engine B, the models are fundamentally processing different starting sequences, which alters the draft predictions and invalidates an apples-to-apples comparison of the speculation algorithm itself. We aim to eliminate this external noise, ensuring the exact same initial token IDs are fed to the models.
> * **Internal factors:** We consider numerical differences, kernel optimizations, and continuous batching mechanisms to be core components of an engine's internal logic. We **do not** want to mask these differences. On the contrary, we *want to measure them end-to-end*. The goal is to see how the speculative decoding algorithm behaves when subjected to the actual computational reality of that specific serving framework.
>
> To illustrate why we designed the SPEED-Bench framework to handle tokenization externally, here is a concrete example of differing default behaviors and framework issues that motivated our choice:
> * **vLLM:** An issue across multiple versions has been the inconsistent injection or duplication of Beginning of Sequence (BOS) tokens, as documented in broader tracking issues (e.g., Issue #9519: "Multiple inconsistencies wrt BOS injection") and specific bug reports for their chat pipelines (Issue #2012, Issue #16853, PR #16081).
> * **SGLang:** Similar chat template and API adapter bugs alter the prompt sequence. For example, Issue #3728 reported an accidental additional <|begin_of_sentence|> token being injected for DeepSeek R1 right before the model was supposed to generate an answer, and PR #3432 was recently needed to fix BOS handling in their OpenAI API adapter.
> * Even without bugs, engines apply tokenizer defaults differently. We ran a simple test sending the exact same chat payload `({"content": "<|startoftext|>I am going to Paris, what should I see?", "role": "user"})` for a GPT-OSS model to both TensorRT-LLM (v1.3.0rc9) and vLLM (v0.16.0). TensorRT-LLM sets `add_special_tokens=True` by default, yielding a 89-token sequence with injected control tokens. Conversely, vLLM defaults to `add_special_tokens=False`, yielding a 83-token sequence. If SPEED-Bench did not handle this, the draft models in these two engines would be conditioned on different contexts.
>
> Finally, we emphasize that beyond prompt handling, a large contribution of our measurement framework is the effort required to bridge these systems. By standardizing the *client-side dispatch and metrics collection*, SPEED-Bench allows practitioners to easily run and evaluate the exact same realistic workloads across entirely different production engines, which we believe provides  value to the community.
>
> We will add a paragraph to Section 7 clarifying this distinction between internal and external factors.

---

> > ### Author Rebuttal · Reviewer_NJCC · 2026-04-03
> >
> > Thanks for the detailed explanations. I will keep my positive score to recommend acceptance.

---

> > > ### Author Response · Authors · 2026-04-05
> > >
> > > Dear reviewer,
> > >
> > > Thank you again for your time, your constructive feedback, and for acknowledging that our rebuttal resolved your concerns. We truly appreciate your positive recommendation for the paper's acceptance.
> > >
> > > Since the technical weaknesses have been addressed, we wanted to humbly ask if you might be open to raising the numerical score. Alternatively, if there are any remaining suggestions or areas for improvement that we might have missed that would help elevate the paper further, we would be more than happy to address them.
> > > We are very grateful for your engagement and for helping us strengthen our work.
> > >
> > > Best,
> > > The Authors

---

### Official Review · Reviewer_4fmb · 2026-03-22

**Soundness:** 3
**Presentation:** 4
**Significance:** 3
**Originality:** 3
**Overall Recommendation:** 5
**Confidence:** 4

**Summary:**

SPEED-Bench proposes a more realistic benchmark for speculative decoding by combining a semantically diverse Qualitative Split, a serving-oriented Throughput Split, and a unified measurement framework compatible with production engines such as vLLM, TensorRT-LLM, and SGLang. The paper’s main argument is that speculative decoding performance depends heavily on workload diversity, input length, and batch regime, so common evaluations based on small, redundant datasets, batch size 1, or synthetic random-token inputs can misrepresent real deployment behavior. The benchmark expands coverage versus SpecBench with 11 categories, 24 source datasets, longer prompts, multilingual breadth, and explicit long-context / large-batch throughput testing from 1k to 32k input lengths. Using this setup, the authors show that semantic diversity changes conclusions, random-token benchmarking is unreliable, optimal draft length shifts with concurrency, and long-context drafter accuracy can degrade beyond training length but can be partly recovered with techniques like YaRN scaling.

**Compliance With Llm Reviewing Policy:**

Affirmed.

**Key Questions For Authors:**

Thank you for submitting to ICML! I like the benchmark paper. I hope the authors can have more insights from the benchmark.

**Strengths And Weaknesses:**

## Strengths and Weakness
+ clear writing
+ strong motivation and solve an important problem
+ detailed and useful information in appendix
- limited SD methods
- takeaways are limited
- using input of fixed size in throughput measurement without considering how the mix of prefill and decode affect throughput

---

> ### Author Rebuttal · Authors · 2026-03-29
>
> We thank the reviewer for their positive feedback and for recognizing the importance of our proposed benchmark. We address your specific concerns below:
>
> * *limited SD methods* - We designed the SPEED-Bench framework not as a standalone tool, but as a lightweight “wrapper” that handles the data, tokenization, and statistic gathering externally while integrating directly with production-grade engines like SGLang, vLLM, and TensorRT-LLM. Because of this design, our benchmark is method-agnostic and supports any SD algorithm implemented in the underlying backends. While our primary experiments focused on draft chains (the standard for $BS>1$ speedups in production), the framework supports tree-based evaluation as well. To demonstrate this flexibility, we ran additional evaluations using tree-based speculation for Qwen3 235B with Eagle3 in SGLang. In these experiments, we vary the branching factor (top-k) from 2 to 8 and the draft length (DL) from 3 to 7, as summarized in the table below. We will include these results, along with additional experiments on tree-based methods, in the final version of the paper.
> |top-k\DL|3|5|7|
> |---|---|---|---|
> |2|2.55|2.81|2.89|
> |4|2.82|3.19|3.31|
> |8|3.04|3.49|3.65|
>
> * *takeaways are limited* - We respectfully highlight that SPEED-Bench exposes a few critical system behaviors that are often masked by traditional evaluation methods:
>    * Using random token batches artificially skews acceptance rates and overestimates real-world throughput by an average of 23%. Furthermore, random noise fails to trigger realistic expert routing in Mixture-of-Experts (MoE) models, causing routers to collapse to a subset of experts and invalidating step latency measurements.
>    * We show that aggressive optimizations, like pruning drafter vocabularies to 32k tokens in Eagle3, severely degrade performance on the "long tail" of user inputs, particularly harming the Multilingual, RAG, and Summarization domains. The RAG and Summarization domains require repeating input text, thus having the specific tokens dropped from the vocabulary can degrade results dramatically.
>    * Through our fixed Input Sequence Length (ISL) buckets, we reveal that drafter accuracy rapidly degrades when the inference ISL exceeds the training ISL. However, we also demonstrate that applying YaRN scaling at inference time can recover significant drafting accuracy.
>    * We empirically show that the optimal DL is highly sensitive to the concurrency regime. Memory-bound scenarios (low batch sizes) benefit from longer drafts, while compute-bound scenarios (high batch sizes) shift the optimal configuration closer to a single draft token.
>
> * *using input of fixed size in throughput measurement without considering how the mix of prefill and decode affect throughput* - This is an excellent observation. We agree that real-world production workloads feature a complex mix of prefill and decode phases. We deliberately constructed the Throughput Split into fixed ISL buckets to isolate the specific performance effects of context length and to construct stable throughput-latency Pareto curves. However, because our measurement framework operates asynchronously, it concurrently dispatches these requests to engines like vLLM and TensorRT-LLM. Because these engines utilize continuous batching under the hood, *they are natively handling a complex mix of prefill and decode operations during the benchmark itself*. Furthermore, if practitioners specifically wish to benchmark a natural, highly heterogeneous mix of sequence lengths, they can utilize our Qualitative Split. As detailed in Appendix A (Table 3), this split contains a wide natural variance of ISLs across its 11 categories. We will add a clarifying note to Section 6/7 to better highlight how our dataset splits and asynchronous framework capture this behavior.

---

> > ### Author Rebuttal · Reviewer_4fmb · 2026-03-31
> >
> > Thank you for the response!

---

### Decision · Program_Chairs · 2026-04-30

**Decision:**

Accept (regular)

**Comment:**

This paper proposes SPEED-Bench, a standardized benchmark for evaluating speculative decoding algorithms and a unified measurement framework integrated with major serving engines. Key findings include that synthetic benchmarks substantially overestimate real-world throughput and that vocabulary pruning in existing methods degrades multilingual acceptance length.

The rebuttal was particularly strong, providing new evidence for selection algorithm stability and ablation of the sample selection method. This is a timely and important contribution that addresses a real evaluation gap in the speculative decoding community. The paper is well-motivated, practically valuable, and the experimental methodology is thorough.